EMBO
Molecular Medicine

# A paracrine activin A–mDia2 axis promotes squamous carcinogenesis via fibroblast reprogramming

Michael Cangkrama[1],* , Mateusz Wietecha[1], Nicolas Mathis[1], Rin Okumura[1], Luca Ferrarese[1], Dunja Al-Nuaimi[1], Maria Antsiferova[1,†], Reinhard Dummer[2], Metello Innocenti[3] & Sabine Werner[1],**

## Abstract

Cancer-associated fibroblasts (CAFs) are key regulators of tumorigenesis and promising targets for next-generation therapies. We discovered that cancer cell-derived activin A reprograms fibroblasts into pro-tumorigenic CAFs. Mechanistically, this occurs via Smad2-mediated transcriptional regulation of the formin mDia2, which directly promotes filopodia formation and cell migration. mDia2 also induces expression of CAF marker genes through prevention of p53 nuclear accumulation, resulting in the production of a pro-tumorigenic matrisome and secretome. The translational relevance of this finding is reflected by activin A overexpression in tumor cells and of mDia2 in the stroma of skin cancer and other malignancies and the correlation of high activin A/mDia2 levels with poor patient survival. Blockade of this signaling axis using inhibitors of activin, activin receptors, or mDia2 suppressed cancer cell malignancy and squamous carcinogenesis in 3D organotypic cultures, *ex vivo*, and *in vivo*, providing a rationale for pharmacological inhibition of activin A-mDia2 signaling in stratified cancer patients.

**Keywords** activin; CAF; carcinogenesis; mDia2; tumor microenvironment
**Subject Categories** Cancer; Skin

See also: **R Samain & V Sanz-Moreno** (April 2020)

## Introduction

Skin cancer is by far the most common type of cancer worldwide, and non-melanoma skin cancers (NMSC) are particularly common (Lomas *et al*, 2012). About 5.4 million cases of NMSCs, which include basal cell carcinoma (BCC), cutaneous squamous cell carcinoma (cSCC), and a few less common malignancies, were treated in the United States in 2012, and their incidence is increasing at an alarming rate (Rogers *et al*, 2015). This is mainly a consequence of the increased life expectancy, the extended sun exposure, and the use of immunosuppressive therapy in organ transplant patients (Lanz *et al*, 2019). Therefore, there is a strong need to unravel the molecular drivers of these tumors and to develop new strategies for efficient prevention and treatment of NMSCs.

In addition to mutations and epigenetic changes in the tumor cells, alterations in stromal cells strongly contribute to carcinogenesis (Dotto, 2014; Curtius *et al*, 2018). Therefore, treatments targeting different components of the tumor microenvironment (TME), such as the tumor's blood and lymphatic vasculature as well as cancer-associated immune cells, have been proposed (Binnewies *et al*, 2018). A particularly important component of the TME are CAFs (Kalluri, 2016). They reside at the tumor margin or infiltrate the tumor, and affect cancer development by secreting various growth factors, cytokines, chemokines, proteinases, and extracellular matrix (ECM) proteins (Madar *et al*, 2013). These features of CAFs are relatively well characterized and in most cases pro-tumorigenic (Gascard & Tlsty, 2016). However, the factors responsible for the induction of a CAF phenotype in different cancers and their molecular targets are still understudied.

Here, we identified activin A as a major player in CAF differentiation. This transforming growth factor β (TGF-β) family member is mainly known as a reproductive hormone, but potent activities of activin A in tissue repair and fibrosis have also been discovered (Werner & Alzheimer, 2006; Hedger & de Kretser, 2013). Consistent with the remarkable parallels between tissue repair and cancer (Schafer & Werner, 2008), important roles of activin A in different malignancies are emerging (Antsiferova & Werner, 2012), although the underlying mechanisms are still largely unexplored. Interestingly, expression of activin A is strongly upregulated in human BCCs and cSCCs (Antsiferova *et al*, 2011). This is functionally relevant, since we previously showed that overexpression of activin A in keratinocytes

1 Department of Biology, Institute of Molecular Health Sciences, ETH Zurich, Zurich, Switzerland
2 Department of Dermatology, University Hospital Zurich, Zurich, Switzerland
3 Heidelberg University Biochemistry Center (BZH), Heidelberg University, Heidelberg, Germany
*Corresponding author. Tel: +41 44 633 3941; Fax: +41 44 633 1174; E-mail: michael.cangkrama@biol.ethz.ch
**Corresponding author. Tel: +41 44 633 3941; Fax: +41 44 633 1174; E-mail: sabine.werner@biol.ethz.ch
†Present address: Roche Glycart AG, Schlieren, Switzerland

of transgenic mice to a similar extent as in human SCCs promoted formation of skin papillomas and their malignant progression in a non-cell-autonomous manner (Antsiferova *et al*, 2011). However, genetic or pharmacological depletion of different immune cells did not or only partially suppress the activin A-induced promotion of squamous carcinogenesis (Antsiferova *et al*, 2013, 2017), indicating key roles of other cells of the TME in the development and progression of NMSC. Most importantly, the molecular targets of activin A that drive tumor development and progression are as yet unknown.

Here, we show that activin A induces a pro-tumorigenic CAF phenotype in skin fibroblasts through activation of a Smad2-mDia2-p53 signaling axis. This unexpected link between major regulators of fibroblast function and tumorigenesis offers promising therapeutic opportunities for the treatment of different malignancies.

## Results

### Activin A promotes SCC cell malignancy in a non-cell-autonomous manner

To determine the potential involvement of non-immune cells in the pro-tumorigenic activity of activin A, we used a xenograft model of immunodeficient NOD/SCID mice. Weakly tumorigenic cSCC cells (SCC13), which express low levels of endogenous activin A, were transduced with lentiviruses expressing the inhibin/activin βA subunit (*INHBA*) in a doxycycline (DOX)-inducible manner (SCC13 Act) (Fig 1A). This mimics the situation in human NMSC, where INHBA is up to 50-fold overexpressed in total tumor samples compared to normal skin and even more in the cancer cells themselves (Antsiferova *et al*, 2011). In DOX-treated SCC13 Act cells, *INHBA* mRNA levels were approximately 70- to 90-fold increased, and activin βA precursor and mature protein were detected in the lysate or medium, respectively

(Fig 1B). Expression of *INHBB* or *INHA* was not induced, suggesting that overexpression of *INHBA* results mainly in production of activin A. Expression of *TGFB1* and of the secreted activin antagonist follistatin (*FST*) was also not affected (Fig EV1A).

Activin A overexpression did not affect proliferation, migration, or the epithelial phenotype of SCC13 cells in spite of efficient nuclear translocation of SMAD2/3 (Fig EV1B and C). However, upon intradermal injection of SCC13 cells or of highly malignant A431 epidermoid vulvar carcinoma cells into ears of NOD/SCID mice, the tumors formed by activin A-overexpressing cells were larger compared to those formed by empty vector (EV)-transduced cells (SCC13 EV, A431 EV) (Figs 1C–E and EV1D–H) and showed invasive growth as revealed by penetration through the ear cartilage (Fig 1D, inset). The cancer cells still expressed E-cadherin, but also showed cytoplasmic/nuclear β-catenin (Fig EV1I). Areas with high collagen density were significantly enlarged in these tumors, indicating a pro-fibrotic fibroblast phenotype (Fig 1F). Consistent with this finding, the proliferation rate of stromal fibroblasts was significantly increased (Fig 1G). There was no change or even a slight decrease in the area covered by blood or lymphatic vessels (Fig EV1J and K), which is consistent with the anti-angiogenic/anti-lymphangiogenic role of activin A (Krneta *et al*, 2006; Kaneda *et al*, 2011; Heinz *et al*, 2015). Overall, these findings point to a role of fibroblasts in the pro-tumorigenic effect of activin A.

### Activin A promotes SCC cell malignancy in organotypic cultures

To further test this possibility, we established 3D organotypic cultures of cancer cells with primary human dermal fibroblasts. Remarkably, the epithelium formed by SCC13 Act cells was hyperthickened and disorganized (Fig 1H and I), and the tumor cells invaded through the basement membrane into the dermal equivalent (Fig 1J). This was also observed with A431 cells and with SCC13 Act cells in co-culture

---

**Figure 1.  Activin A promotes squamous cell malignancy and tumor growth in a xenograft model and in organotypic culture.**

A  qRT–PCR for *INHBA* relative to *RPL27* using RNA from SCC13 cells transduced with a lentiviral vector allowing expression of *INHBA* in a doxycycline (DOX)-inducible manner (SCC13 Act clone 1 and 2) or empty vector (EV) (N = 3). SCC13 Act and EV cell lines were generated from lentivirally transduced cells upon single-cell clonal expansion.

B  Western blot of cell lysate and conditioned media (CM; bottom) of transduced SCC13 cells for the activin βA subunit and GAPDH (loading control for cell lysate) under reducing conditions. The higher molecular weight of recombinant activin βA results from the HA-epitope tag. Ponceau S staining of the membrane was used as a loading control for the CM.

C  Representative pictures of 5-week-old tumors (indicated by arrows) formed in NOD/SCID mice upon intradermal injection of SCC13 Act (clone 2) or SCC13 EV cells and tumor volume at different time points of tumor development. N = 3–6 tumors.

D  Representative images of H&E stainings of tumors formed by SCC13 EV and Act cells (clone 2) at day 35 (asterisk indicates cartilage; arrow indicates site of invasion). An inset panel indicates the area where tumor cells invade into the cartilage. Tumor sections were evaluated for invasion through the basement membrane. N = 6 (EV), N = 6 Act (3 tumors from clone 1 and 3 tumors from clone 2). Tumors from Act mice were pooled in the graph.

E  Tumor weight at endpoint (5 weeks after injection). N = 3–6 tumors.

F  Herovici staining and quantification of percentage of collagen-positive (dark blue) area per field of view. N = 3 tumors, n = 3–4 histological sections.

G  Representative images of sections from tumors formed by SCC13 EV and SCC13 Act cells (clone 2) stained for Ki67 (red), PDGFR-α (green), and counterstained with Hoechst (blue). Quantification of Ki67/PDGFR-α-positive cells is shown in the graph. N = 3 tumors, n = 3–4 histological sections.

H  Representative images of sections from 3D cultures of primary human skin fibroblasts with either SCC13 EV or SCC13 Act (clones 1 and 2) cells stained with either H&E or Herovici.

I  Quantification of Ki67-positive cells in sections from SCC13 EV and SCC13 Act (clone 2) organotypic cultures (co-stained with E-cadherin to indicate epithelial cells). N = 3 independent cultures, n = 3 histological sections.

J  Representative images of sections from SCC13 EV and SCC13 Act (clone 2) organotypic cultures stained for K14 (red) and collagen IV (green) combined with Hoechst staining (blue) and quantification of the K14-positive area below the basement membrane (BM). N = 3 independent cultures, n = 4 histological sections. The area indicated with a white rectangle in the left picture is shown at higher magnification in the right picture.

Data information: Bar graphs show mean ± SEM. *P < 0.05, **P < 0.01, ***P < 0.001, ****P < 0.0001 (one-way ANOVA with Bonferroni post-test (A, J), two-way ANOVA with Bonferroni post hoc test (C, I), unpaired Student's *t*-test (E (combined), F, G), Fisher's exact test (D)). Scale bars, 500 μm (D) or 100 μm (F–H, J). Exact P-values are provided in Dataset EV3.

Source data are available online for this figure.

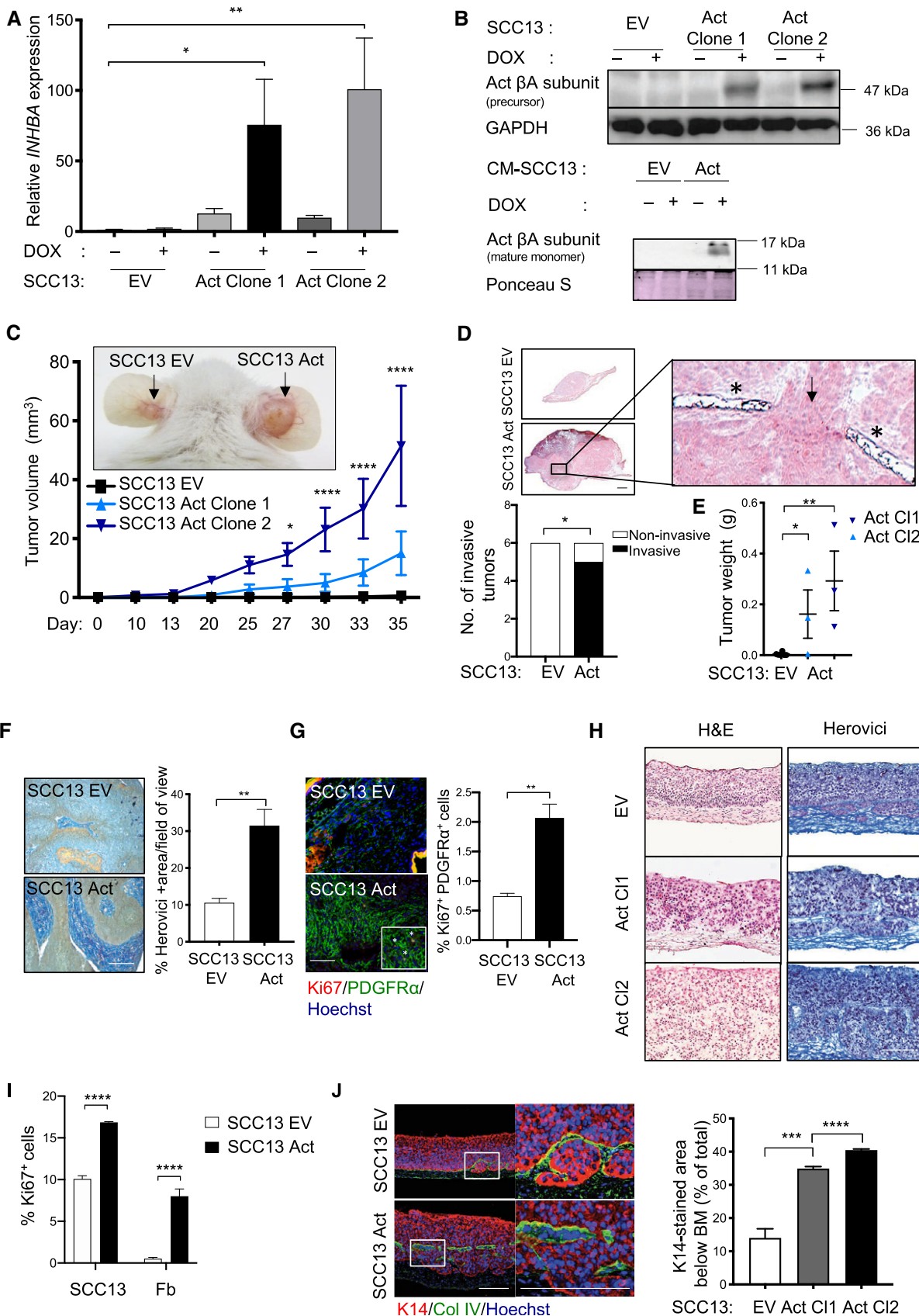

**Figure 1.**

with either primary or immortalized mouse fibroblasts (Fig EV2A and B). Activin A-overexpressing cells showed various malignant features, including reduced membrane staining for E-cadherin in cells at the invasive front, cytoplasmic/nuclear β-catenin, and increased secretion of active matrix metalloproteinases (MMP)-2 and MMP-9 (Fig EV2C–E). Since these data show that activin A promotes invasive growth of SCC cells via fibroblasts, we next determined whether activin A directly affects these cells.

### Activin A induces a CAF phenotype in skin fibroblasts

Recombinant activin A promoted proliferation of primary dermal fibroblasts in a dose-dependent manner (Fig 2A). However, it did not stimulate contraction of collagen gels by primary mouse and human fibroblasts, while TGF-β1—a key regulator of myofibroblast differentiation (Tomasek *et al*, 2002)—had a strong effect (Fig 2B; Appendix Fig S1A). Activin A even inhibited this activity of TGF-β1 (Fig 2B). Although both activin A and TGF-β1 increased the expression of alpha smooth muscle actin (αSMA, encoded by the *Acta2* gene), a marker for contractile myofibroblasts (Tomasek *et al*, 2002) (Fig 2C; Appendix Fig S1B), only the latter showed incorporation into stress fibers (Fig 2D), a characteristic feature of myofibroblasts (Clement *et al*, 2005). On the contrary, only activin A enhanced the density and length of actin-rich filopodia and promoted migration in scratch and chemotactic transwell assays (Fig 2E–G). Since high motility is a hallmark of CAFs, including cultured CAFs from human cSSCs (Commandeur *et al*, 2011), and implicated in cancer cell invasion (Karagiannis *et al*, 2012; Kalluri, 2016), we determined whether activin A also induces the expression of CAF marker genes in normal fibroblasts. Indeed, several skin CAF marker genes, which encode predominantly cytokines and ECM proteins (Procopio *et al*, 2015), were upregulated in activin A-treated fibroblasts. Expression of endogenous *Inhba* was also increased, indicating autoinduction (Fig 2H).

### Activin A stimulates expression and release of tumor-promoting factors in fibroblasts

To investigate the effect of activin A on the secretome and matrisome of fibroblasts and the relevance of these secreted factors for cancer cell behavior, we generated human primary fibroblasts with DOX-inducible overexpression of activin A (Fb Act), thus mimicking the activin A autoinduction seen upon exposure of fibroblasts to exogenous activin A. We selected clones, which show a fourfold to fivefold *INHBA* overexpression that is comparable to the overexpression seen in response to activin A treatment (Fig 3A and B). These cells produced a secretome, which promoted migration and clonogenicity of cancer cells and deposited increased levels of fibronectin and collagen I (Fig 3C–E). To determine whether the deposited matrix has pro-tumorigenic activities, we plated SCC13 cells on the de-cellularized matrix deposited by either Fb Act or Fb EV cells. Indeed, the colony-forming and migratory capacities of the cancer cells were significantly higher on matrix deposited by activin-overexpressing cells. A similar effect was seen in direct 2D co-culture (Fig 3F–H). However, it was less pronounced, since ECM and conditioned medium were collected for 3 days and the conditioned medium was concentrated. The conditioned medium of Fb Act also promoted anchorage-independent growth of SCC13 cells as shown in a spheroid formation assay. However, recombinant activin A alone had no effect in this assay (Fig 3I), suggesting that other factors secreted by these cells, but not activin A itself, enhance proliferation and invasive growth of cancer cells. Consistently, expansion of SCC13 cells in 3D cultures was significantly increased when the cancer cells were seeded on a dermal equivalent formed by activin A-overexpressing fibroblasts (Fig 3J).

Most importantly, when SCC13 cells were co-injected with Fb Act or EV, tumor formation was strongly accelerated upon DOX-induced *INHBA* expression (Fig 3K). These data demonstrate that activin A induces the production of secretomes and matrisomes by fibroblasts, which promote tumor cell proliferation and invasive growth.

### Activin A induces CAF gene expression by fibroblasts *in vivo*

To unravel the molecular mechanisms underlying this effect, we used a transgenic mouse model where skin papillomas form spontaneously due to expression of human papilloma virus 8 (HPV8) oncogenes in keratinocytes (Schaper *et al*, 2005). When these mice (HPV8/wt mice) were mated with transgenic mice overexpressing *INHBA* in keratinocytes (wt/Act mice), HPV8-induced tumor

---

**Figure 2. Activin A induces a CAF phenotype in fibroblasts.**

A  Primary murine dermal fibroblasts, which had been treated with recombinant activin A at different concentrations, were analyzed for BrdU incorporation. N = 3–4.

B  Representative images of collagen gels and quantification of the extent of gel contraction (percentage of initial gel size) in the presence or absence of activin A (20 ng/ml) and/or TGF-β1 (1 ng/ml). N = 3–4.

C  qRT–PCR and Western blot analyses of *ACTA2*/αSMA expression using total RNA/lysates of fibroblasts treated with activin A or TGF-β1 for 6 h. N = 3. The membrane was re-probed with GAPDH and mDia2 antibodies.

D  Representative images of fibroblasts treated with activin A, TGF-β1, or vehicle (control) as in (C), stained for αSMA (green), counterstained with rhodamine-coupled phalloidin (red), and Hoechst (blue). Bar graph shows percentage of fibroblasts with stress fibers. N = 3.

E  Representative images of control and activin A-treated fibroblasts stained with rhodamine-coupled phalloidin; scatter plot shows quantification of filopodium length in control, activin A-treated, and TGF-β1-treated fibroblasts. N = 27–70 filopodia.

F  Scratch assay of fibroblasts treated with activin A or vehicle for 24 h. N = 3. The area covered by cells in the original scratch was quantified. Migration in one control culture was set to 1.

G  Chemotactic migration of primary mouse fibroblasts from wt mice for 24 h in a transwell assay toward increasing concentrations of activin A or vehicle. N = 3.

H  qRT–PCR analysis of CAF marker genes relative to *Rps29* using RNA from fibroblasts treated with activin A for 6 h. N = 3.

Data information: Bar graphs show mean ± SEM. ns $P > 0.05$, *$P < 0.05$, **$P < 0.01$, ***$P < 0.001$ ****$P < 0.0001$ (one-way ANOVA with Bonferroni post-test (A–G) or two-way ANOVA with Bonferroni post hoc test (H)). Scale bars: 50 μm (D) and 25 μm (E). Exact *P*-values are provided in Dataset EV3.
Source data are available online for this figure.

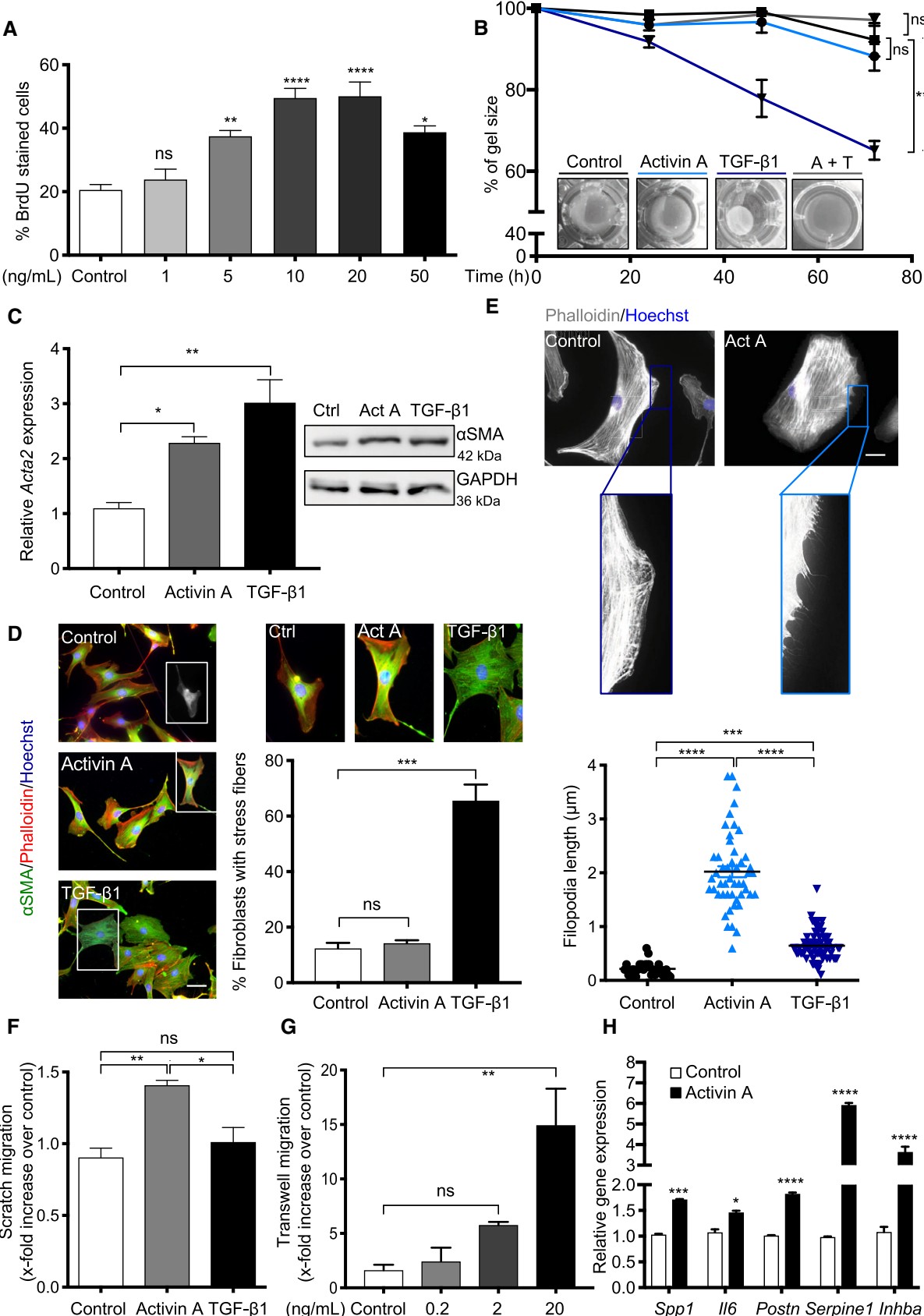

Figure 2.

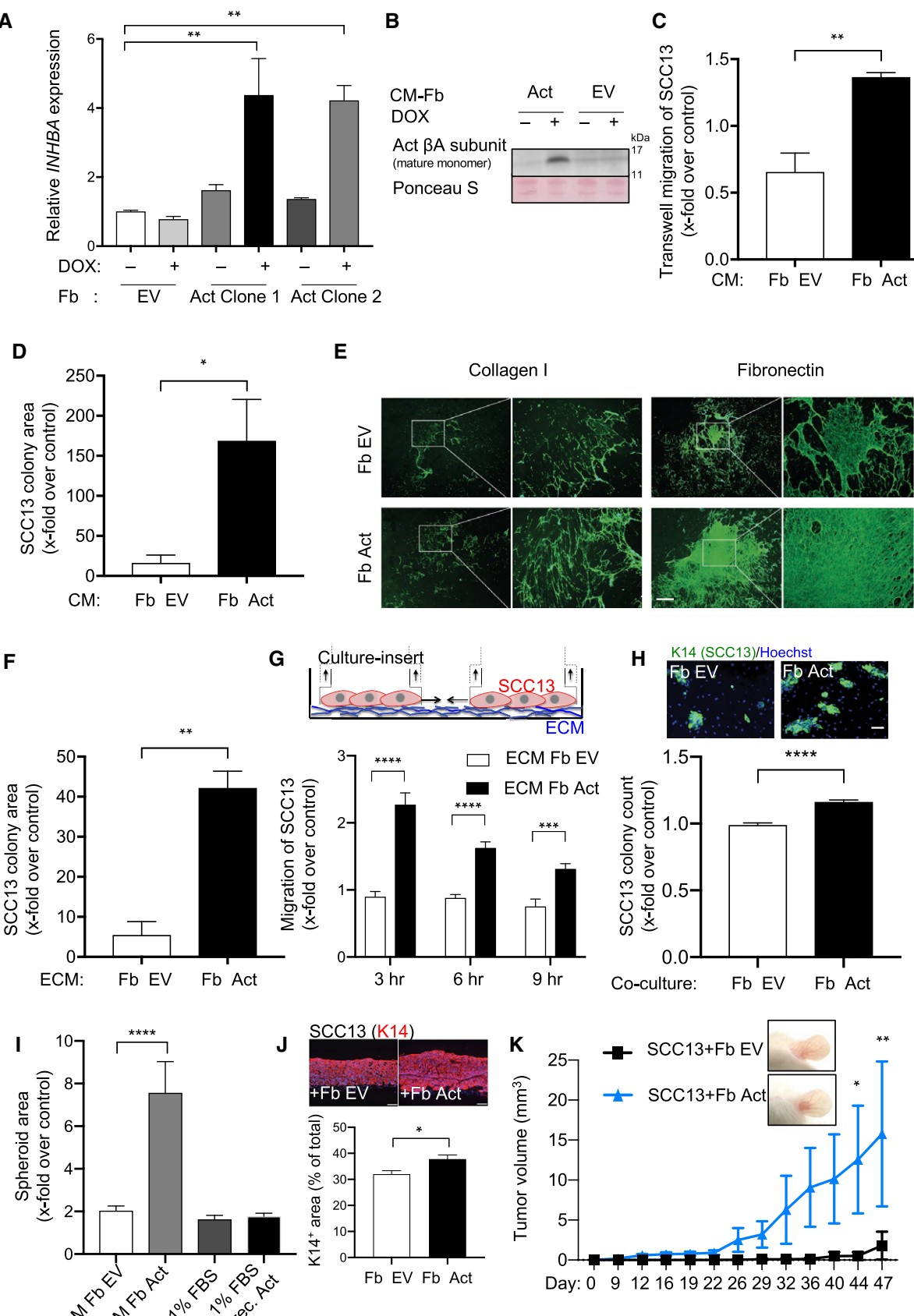

Figure 3.

**Figure 3.  Activin A-exposed fibroblasts produce a tumor-promoting secretome and matrisome.**

A   qRT−PCR for *INHBA* relative to *RPL27* using RNA from primary human fibroblasts transduced with a lentiviral vector allowing expression of *INHBA* in an inducible manner after treatment with DOX for 24 h (Fb Act, clones 1 and 2) or with empty vector (Fb EV) (N = 3). Fb Act and Fb EV cultures were generated from lentivirally transduced cells upon clonal expansion of resistant single cells.

B   Western blot of conditioned media (CM) of transduced fibroblasts showing mature activin βA. Ponceau S staining of the membrane was used as a loading control.

C   Chemotactic migration of SCC13 cells for 24 h in a transwell assay containing CM from Fb Act (clone 2) or Fb EV. N = 4. Migration in one control culture was set to 1.

D   Relative colony area of SCC13 cells upon culture in CM from Fb Act (clone 2) or EV. N = 3.

E   Representative immunofluorescence images of de-cellularized matrix from Fb Act (clone 2) and Fb EV stained for fibronectin or collagen I.

F   Relative colony area of SCC13 cells plated on either de-cellularized matrix from Fb Act (clone 2) or Fb EV. N = 3.

G   Migration of SCC13 cells plated on either de-cellularized matrix from Fb Act (clone 2) or Fb EV. N = 6. The experimental setup is shown schematically above the graph.

H   Fb Act (clone 2) or Fb EV cells were cultured together in 2D with SCC13 cells. After 7 days of culture, SCC13 cells were identified by K14 staining and the number of K14-positive colonies was quantified. N = 6.

I   Quantification of SCC13 tumor spheroid area in single hanging drop including CM from Fb EV or Fb Act (clone 2) (including 1% FBS), DMEM/1% FBS, or DMEM/1% FBS plus 20 ng/ml recombinant activin A. The area of the spheroid formed in one Fb EV sample was set to 1. N = 9–10 hanging drops.

J   Representative images and quantification of K14-stained areas from 3D cultures of SCC13 cells seeded on top of either Fb Act (clone 2) or Fb EV. N = 3 independent cultures, n = 3 histological sections.

K   Representative pictures of tumors formed in NOD/SCID mice 6–7 weeks after intradermal injection of SCC13 cells with Fb Act (clone 2) or Fb EV. Tumor volume at different time points of tumor development (6–7 weeks after injection). N = 5 tumors per group.

Data information: Bar graphs show mean ± SEM. *P < 0.05, **P < 0.01, ***P < 0.001, ****P < 0.0001 [one-way ANOVA with Bonferroni post-test (A, I), two-way ANOVA with Bonferroni post hoc test (K), multiple Student's *t*-test (G), unpaired Student's *t*-test (C, D, F, H)]. Scale bars, 500 μm (E), 50 μm (H), or 100 μm (J). Exact *P*-values are provided in Dataset EV3.

Source data are available online for this figure.

formation was strongly accelerated (Antsiferova *et al*, 2017). We then FACS-isolated fibroblasts from the ear skin of wild-type (wt/wt), wt/Act, HPV8/wt, and HPV8/Act mice prior to the formation of tumors and analyzed their expression profile by RNA sequencing. The purity of this cell fraction was above 90% as determined by re-analysis of the sorted cells (Fig EV3A and B). The number of cells expressing the pan-fibroblast marker PDGFRα (CD140a) (Driskell *et al*, 2013) and concomitantly lacking the immune cell marker CD45 was significantly increased in mice expressing the *INHBA* transgene, and there was a virtual absence of markers for keratinocytes and endothelial and immune cells in the sorted cells (Fig EV3C and D). Expression of the HPV8 transgene in keratinocytes had no major effect on the fibroblast transcriptome, while activin A overexpression in keratinocytes induced major changes in fibroblasts independent of the *HPV8* transgene (Fig EV3E). Gene set enrichment analysis (GSEA) revealed that genes regulated by activin A in fibroblasts showed a significant positive correlation with genes upregulated in human skin SCCs, CAFs from SCCs of patients with recessive dystrophic epidermolysis bullosa (RDEB) (Ng *et al*, 2012), human actinic keratosis (AK), the major SCC precursor lesion (Nindl *et al*, 2006), human breast cancer stroma (Planche *et al*, 2011; Harvell *et al*, 2013) and CAFs (Bauer *et al*, 2010), liver cancer stroma (Sulpice *et al*, 2013), lung cancer stroma and CAFs (Navab *et al*, 2011), and prostate cancer CAFs (Doldi *et al*, 2015), but negative correlation with the genes downregulated in CAFs versus normal tissue fibroblasts (Fig 4A). Interestingly, the gene expression signature of CAFs from Act mice was more similar to the signature of cultured CAFs from the highly aggressive SCCs of RDEB patients than to the signature of regular cSCC CAFs (Fig 4A; Ng *et al*, 2012). Ingenuity pathway analysis (IPA) showed activin A-induced enrichment of genes associated with "proliferation of tumor cells", "migration of cells", and "organization of cytoskeleton" (Fig 4B). Importantly, activin A promoted the expression of various genes encoding ECM components, ECM regulators, and secreted factors (Fig 4C), providing a likely explanation for the pro-tumorigenic activity of the matrisome and secretome of Fb Act cells.

## Activin A transcriptionally activates *mDia2* via Smad2 in fibroblasts

A class of genes that was strongly activated by activin A encodes proteins involved in cell migration, including *mDia2* (*Diap3*) (Figs 4D and EV3E), a member of the formin family of cytoskeletal regulators and key player in filopodium formation (Pellegrin & Mellor, 2005; Beli *et al*, 2008). This is a direct effect, since recombinant activin A induced *mDia2/DIAPH3* expression in primary murine and human fibroblasts (Figs 4E and EV3F). Importantly, TGF-β1 treatment did not increase *mDia2/DIAPH3* expression in skin fibroblasts (Figs 4E and EV3F) or ovarian fibroblasts (Yeung *et al*, 2013). Neither *mDia1* nor *mDia3* expression was induced in Act mice or in activin A-treated cultured fibroblasts (Fig EV3G and H). We identified a conserved "SMAD binding element" (SBE) in the first intron of the *mDia2* gene downstream of the transcription start site (TSS), and activin A, but not TGF-β1, promoted binding of SMAD2/3 to this SBE (Fig 4F). Consistent with the different activity of activin A versus TGF-β1, activin A preferentially activated SMAD2 rather than SMAD3 in other cells (Schmierer *et al*, 2003), and Gene Transcription Regulation Database (GTRD) analysis (Yevshin *et al*, 2017) predicted a higher number of SMAD2 binding sites in the regulatory region of the *mDia2* compared to the *mDia1* and *mDia3* genes (Fig EV3I). The role of activin receptor signaling in the regulation of *mDia2* expression was confirmed by expression of a dominant-negative mutant of activin receptor IB (dnActRIB) in fibroblasts, which reduced basal and activin A-induced mDia2 mRNA levels (Fig 4G).

mDia2 expression correlates with activin A levels in mouse xenograft cancer samples as demonstrated by immunostaining of sections from ear tumors. mDia2 was mainly expressed in the tumor stroma, in particular in cells adjacent to the activin A-secreting tumor cells, which are most likely fibroblasts (Fig 4H), suggesting involvement of mDia2 in the pro-tumorigenic effect of activin A. Furthermore, mDia2 co-localized with the vascular endothelial cell marker cell MECA32, but not with the lymphatic endothelial cell

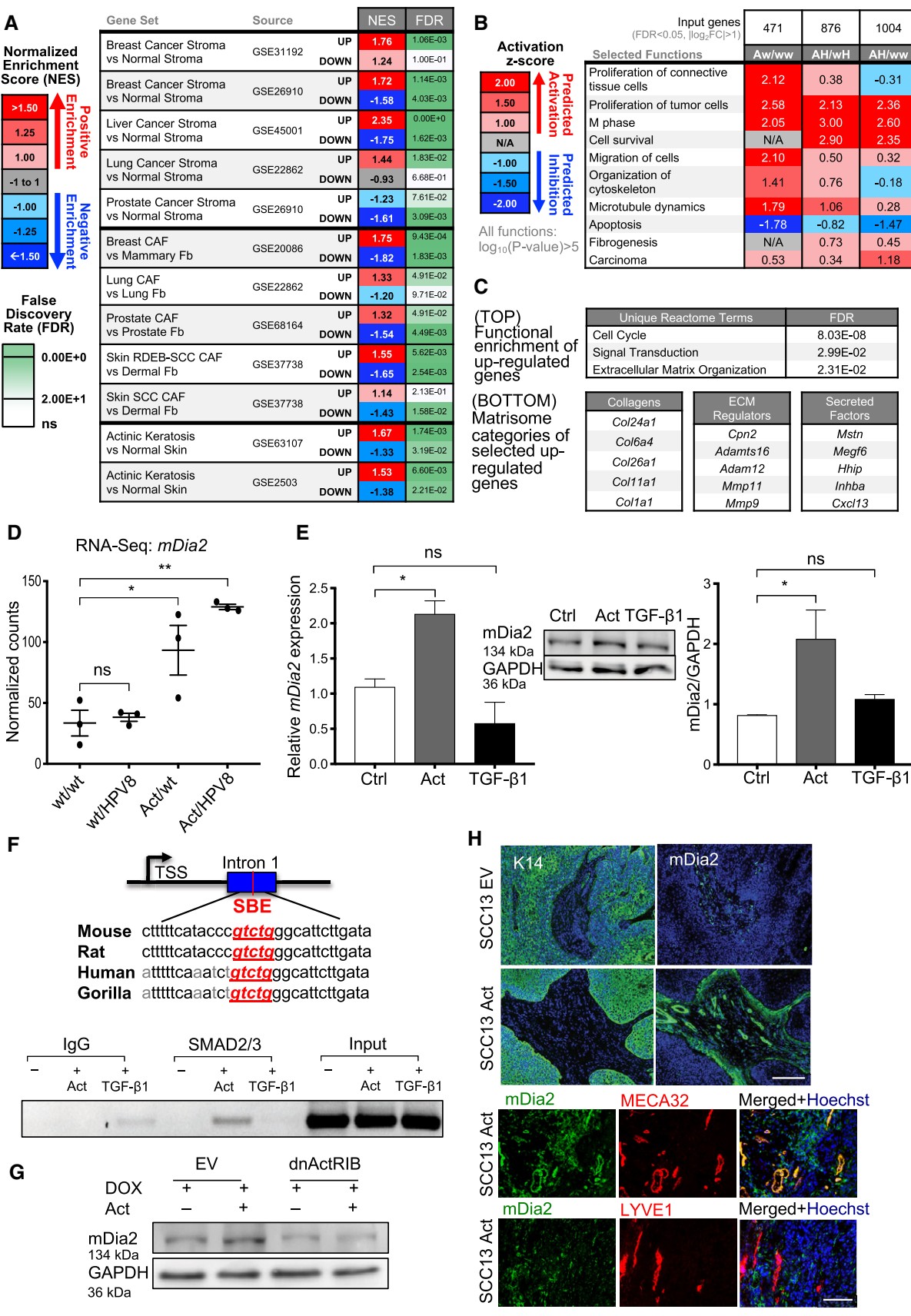

Figure 4.

**Figure 4.  *mDia2* is a direct activin A target gene in fibroblasts.**

A    Gene set enrichment analysis comparing the transcriptomes of fibroblasts isolated from pre-tumorigenic ear skin of female HPV8/Act-transgenic mice versus HPV8/
      wt-transgenic mice (12–14 weeks old) to published datasets of cancer stroma or CAFs from different types of human cancers and from total actinic keratosis (AK)
      samples. Positive or negative normalized enrichment scores (NES) correspond to enrichment of a given set in genes that are up- or downregulated, respectively, in
      response to activin A. Data are presented as pseudo-heatmap with NES magnitude color-coded as indicated in the legend; all non-gray comparisons have
      FDR < 0.10.
B    Ingenuity pathway analysis of differentially regulated genes in FACS-isolated fibroblasts from activin A-transgenic mice in either a HPV8-negative or HPV8-positive
      background compared to respective controls, showing selected top activated functions.
C    (Top) Functional enrichment of the most upregulated genes using the Reactome database showing three unique significantly enriched terms. (Bottom) Filtered list of
      upregulated genes based on the Matrisome database showing selected top upregulated collagens, ECM regulators, and secreted factors.
D    *mDia2* (*Diap3*) expression quantified by RNA sequencing of fibroblasts isolated from ear skin. N = 3 pools of 3–6 mice.
E    Primary mouse fibroblasts were treated with activin A or TGF-β1 or vehicle for 6 h and analyzed by qRT–PCR (N = 3) and Western blot for *mDia2* (*Diap3*) and mDia2
      expression, respectively. The GAPDH loading control is identical to the one in Fig 2C, since the same membrane was probed for mDia2, αSMA, and GAPDH.
F    Alignment of the sequences in the first intron of *mDia2* (*Diap3*) genes of different species that harbor a SMAD2/3 binding element (SBE) (consensus sequence
      highlighted in red and italics) (top). Chromatin immunoprecipitation of lysates from activin A- or TGF-β1-treated fibroblasts using a SMAD2/3 antibody and
      amplification of the bound DNA with mDia2 primers (bottom). Pre-immune serum (IgG) was used as negative control, and the input chromatin is shown. N = 3.
G    Immortalized mouse fibroblasts were transduced with a lentivirus expressing a DOX-inducible dominant-negative ActRIB mutant (dnActRIB) or an empty lentivirus
      (EV), treated with activin A for 2.5 h, and analyzed by Western blot for mDia2 and GAPDH.
H    Representative sections of ear skin tumors formed by SCC13 EV or SCC13 Act cells (clone 2) stained for mDia2 or K14 (green) (top), mDia2 (green) or MECA32/LYVE1
      (red, bottom), and counterstained with Hoechst (blue).

Data information: Bar graphs show mean ± SEM. ns *P* > 0.05, **P* < 0.05, ***P* < 0.01 (one-way ANOVA with Bonferroni post hoc test). Scale bars: 100 μm. Exact *P*-values
are provided in Dataset EV3.
Source data are available online for this figure.

marker LYVE1 in tumor sections, demonstrating that blood vessel endothelial cells in these tumors are also mDia2-positive (Fig 4H).

### mDia2 is overexpressed in human cancer stroma and negatively correlates with survival

To determine the translational relevance of our findings, we analyzed *mDia2/DIAPH3* expression in human skin cancers and found a significant increase in *mDia2/DIAPH3* mRNA levels in human BCC and cSCC samples as compared with normal skin (Fig 5A). Cancer cells in these tumors express high levels of activin A (Antsiferova *et al*, 2011), and mDia2/DIAPH3-positive cells were particularly abundant in the stroma (Fig 5B).

Importantly, *mDia2/DIAPH3* overexpression also occurs in numerous other human cancers as revealed by analysis of datasets from The Cancer Genome Atlas (TCGA) (Fig 5C). Furthermore, GEO dataset analysis revealed that *mDia2/DIAPH3* and *INHBA* were both overexpressed in the stromal compartment of several cancers in comparison with the respective normal tissue and also in breast CAFs compared to normal breast fibroblasts (Fig 5D–G). Since most patients with cSCCs do not die from their tumors, we could not relate the *INHBA/mDia2* expression levels to patient survival. However, Kaplan–Meier survival analysis showed a strong association of high co-expression of *mDia2/DIAPH3* and *INHBA* with decreased survival of patients with liver, stomach, and breast cancer (Fig 5H).

### mDia2 induces a pro-tumorigenic phenotype in skin fibroblasts

To determine the role of mDia2 in skin tumorigenesis, we performed shRNA-mediated knock-down in mouse fibroblasts (Fig 6A). mDia2 silencing caused cell enlargement as previously seen for HeLa cells (Beli *et al*, 2008) and reduced cell proliferation and migration, even in the presence of exogenous activin A (Fig 6B and C). These results were verified with a second shRNA and with primary human fibroblasts (Fig EV4A–E). There was no compensatory upregulation of

*mDia1* or *mDia3*, whereas expression of the genes encoding formin-1 and formin-2 (*Fmn1* and *Fmn2*) was increased (Fig 6D). Interestingly, mDia2 knock-down strongly suppressed the expression of several CAF genes (Fig 6D), particularly those that were upregulated in the presence of activin A (Fig 2H). *Vice versa*, overexpression of murine mDia2, which has ~85% homology to human DIAPH3 and is active in human cells (Isogai *et al*, 2015a, 2016), in normal human fibroblasts induced the expression of CAF markers, including *INHBA* and *DIAPH3* (Fig EV4F). In addition, the matrisome and the secretome of fibroblasts overexpressing mDia2 promoted SCC13 colony formation (Fig EV4G and H).

### mDia2 physically interacts with p53 and promotes CAF marker expression

Previous studies had identified mDia2 in a complex with p53 (Isogai *et al*, 2015a), which plays a non-autonomous tumor-suppressive role in fibroblasts and suppresses CAF gene expression (Addadi *et al*, 2010; Procopio *et al*, 2015). While mRNA levels of p53 and those of the Notch effector CSL/RBP-J, which synergizes with p53 in CAF activation (Procopio *et al*, 2015), were not altered upon *mDia2* silencing, p53 accumulated in the nucleus of mDia2 knock-down fibroblasts (Figs 6E and F, and EV4I–K). Activin A treatment of fibroblasts had the opposite effect and reduced nuclear p53 levels (Fig 6G). The effect of mDia2 on p53 is most likely the consequence of a physical interaction of the endogenous proteins as revealed by co-immunoprecipitation analysis of lysates from primary human fibroblasts. Importantly, activin A treatment of these cells increased the levels of p53 in the mDia2 immunoprecipitate (Fig EV4L). To determine whether p53 nuclear accumulation in mDia2 knock-down cells and repression of CAF markers are functionally linked, we treated these fibroblasts with pifithrin-α (PFT-α), which destabilizes nuclear p53, thereby diminishing p53-dependent transcriptional control (Komarov *et al*, 1999). Indeed, PFT-α reduced the levels of nuclear p53 in mDia2-depleted fibroblasts, and partially or completely rescued the mDia2-mediated downregulation of *Postn* and

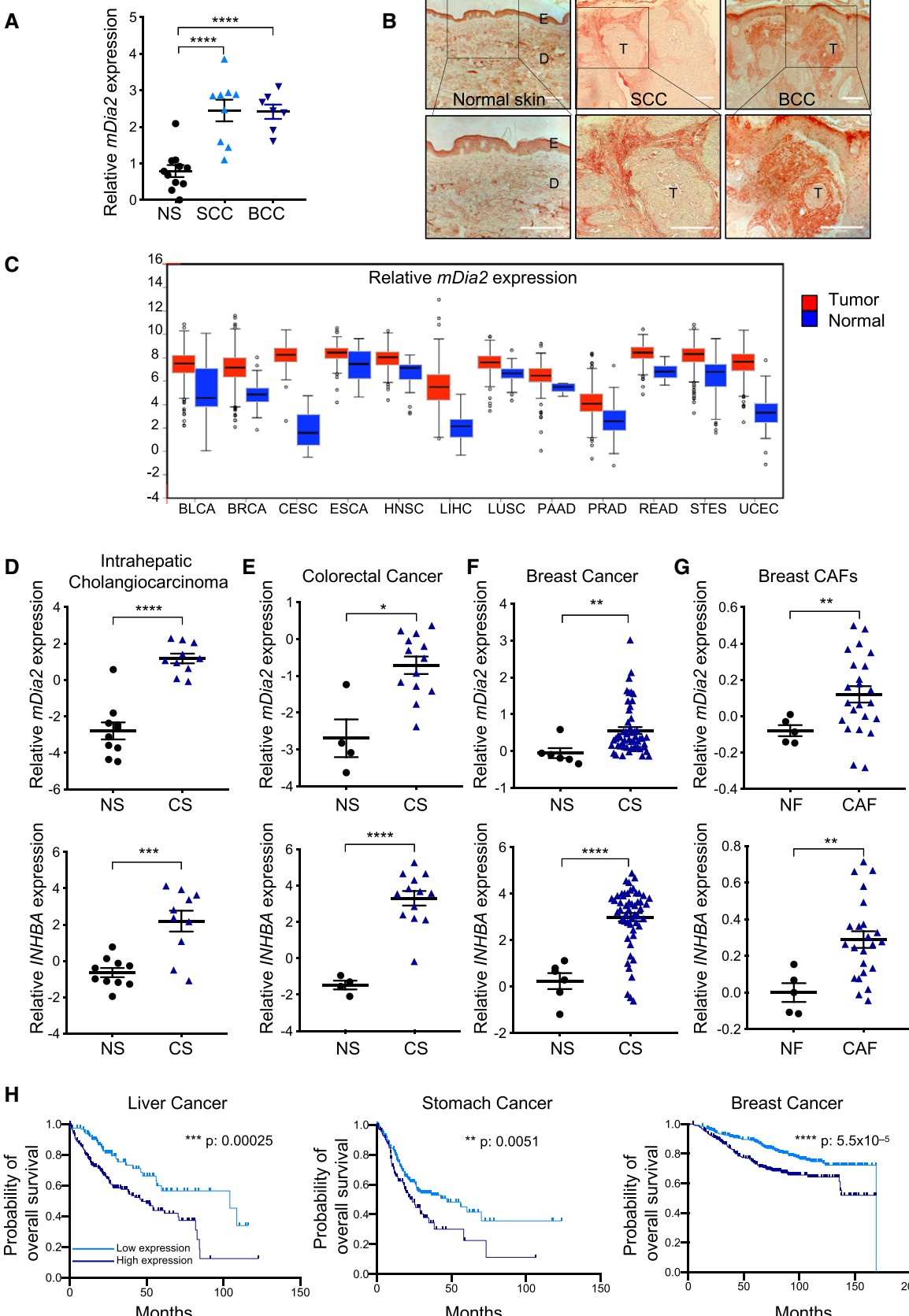

Figure 5.

**Figure 5.  mDia2 (DIAPH3) is overexpressed in human cancers and negatively correlates with survival.**

A  qRT–PCR for *mDia2/DIAPH3* relative to *RPL27* using RNA from normal human skin (*N* = 11), SCCs (*N* = 9), and BCCs (*N* = 7).

B  Representative skin/tumor sections from normal human skin, cSCC, and BCC patients stained for mDia2/DIAPH3. Note the strong mDia2/DIAPH3 positivity in the tumor stroma. Scale bars: 100 μm. D: dermis, E: epidermis: T: tumor.

C  Expression levels of *mDia2/DIAPH3* in tumor and normal tissues based on TCGA cancer data (TCGA data version 20141017, http://firebrowse.org/). Gene expression levels were presented as normalized log2 RSEM (RNA-Seq by Expectation–Maximization) index from the database. BLCA: bladder urothelial carcinoma (*N* = 408 and 19 tumor and normal tissue samples, respectively); BRCA: breast invasive carcinoma (*N* = 1,100 and 112); CESC: cervical SCC (*N* = 306 and 3); ESCA: esophageal SCC (*N* = 185 and 11); HNSC: head-neck SCC (*N* = 522 and 44); LIHC: liver hepatocellular carcinoma (*N* = 373 and 47); LUSC: lung SCC (*N* = 501 and 51); PAAD: pancreatic adenocarcinoma (*N* = 179 and 4); PRAD: prostate adenocarcinoma (*N* = 496 and 50); READ: rectum adenocarcinoma (*N* = 167 and 10); STES: stomach and esophageal carcinoma (*N* = 600 and 46); UCEC: uterine corpus endometrial carcinoma (*N* = 546 and 35). Box plot shows relative *mDia2/DIAPH3* expression in human tumor (red) and normal (blue) tissue. Boxes indicate the interquartile range, whiskers indicate data points within 1.5 times interquartile range, dots outside the whiskers indicate outliers, and the central line indicates median.

D–G  Relative gene expression of *mDia2/DIAPH3* and *INHBA* in the stroma of intrahepatic cholangiocarcinoma, breast and colorectal cancers versus stroma of respective normal tissues based on datasets GSE45001 (D) (*N* = 10 normal tissue & 10 cancer samples), GSE9014 (E) (*N* = 4 and 13), and GSE35602 (F) (*N* = 6 and 53), and in CAFs from breast cancer samples versus normal breast fibroblasts (GSE29270 dataset) (G) (*N* = 5 and 23). NS: normal stroma; CS: cancer stroma; NF: normal fibroblasts; CAF: cancer-associated fibroblasts. The expression values of the datasets were obtained using GEO2R tool in the GEO database.

H  Kaplan–Meier survival curves based on datasets from GEO, EGA, and TCGA, demonstrating that high expression of both *mDia2/DIAPH3* and *INHBA* in patients with liver, stomach, and breast cancer correlates with poor overall survival. Data and statistical analyses were collected from KMPlotter.

Data information: Graphs show mean ± SEM. *P* > 0.05, \**P* < 0.05, \*\**P* < 0.01, \*\*\**P* < 0.001, \*\*\*\**P* < 0.0001, unpaired Student's *t*-test (D–G) and one-way ANOVA with Bonferroni post hoc test (A). Exact *P*-values are provided in Dataset EV3.

*Mmp13*, which are negatively regulated by p53 (Sun *et al*, 2000; Procopio *et al*, 2015), and also of other CAF marker genes (Fig 6H–J). In line with these findings, Ingenuity pathway upstream regulator analysis showed a negative correlation between expression of *INHBA* and p53 target genes in FACS-isolated fibroblasts from Act versus wt mice (Appendix Fig S2A). Overall, these results suggest that activin A induces a CAF phenotype at least in part via mDia2 upregulation and consequent inhibition of p53 nuclear accumulation. Consistent with this assumption, mDia2 knock-down in mouse or human fibroblasts significantly reduced the colony-forming ability of SCC13 Act cells (Figs 6K and EV4M). Most importantly, the efficient tumor formation by SCC13 Act cells that occurred in the presence of control fibroblasts was almost completely abrogated in the presence of mDia2 knock-down fibroblasts (Fig 6L). This correlated with downregulation of periostin and nuclear accumulation of p53 in the tumor stroma (Fig 6M and N). Furthermore, knock-down of mDia2 in INHBA-overexpressing fibroblasts to levels seen in control fibroblasts strongly reduced the positive effect of their secretome and matrisome on SCC13 colony formation (Fig EV4N–P). Taken together, mDia2 is crucial for the activin A-induced conversion of fibroblasts into pro-tumorigenic CAFs and sufficient for induction of a CAF phenotype in normal fibroblasts.

### Repression of the activin A-mDia2 axis inhibits squamous carcinogenesis

Finally, we tested whether blockade of signaling by endogenous activin A inhibits squamous tumorigenesis. To this end, we combined mouse preclinical SCC models with genetic and pharmacological manipulation of the endogenous activin A-mDia2 axis *in vivo* and *ex vivo*.

First, we blocked activin A-induced signaling in fibroblasts through dnActRIB (Appendix Fig S3A and B). Co-injection of SCC13 cells with dnActRIB-expressing fibroblasts delayed tumor growth compared with SCC13 cells co-injected with control fibroblasts, and mDia2-positive cells were significantly less abundant in the stroma (Fig 7A and B). Many of the mDia2-positive cells in the stroma were indeed fibroblasts as revealed by co-staining for mDia2 and the

pan-fibroblast marker PDGFR-α (Fig 7B). Important to note, co-injection of SCC13 cells with normal fibroblasts resulted in a higher rate of tumor formation as compared with injection of SCC13 cells alone (compare Figs 1C and 7A).

Next, we generated SCC13 cells expressing the activin antagonist follistatin (FST-315; Appendix Fig S3C). In the absence of fibroblasts, SCC13 EV cells formed tumors only several weeks after injection. Therefore, we only analyzed the tumors at the day of sacrifice to avoid repeated anesthesia of the mice, which is required for measurement of tumor volume. Importantly, FST overexpression significantly suppressed tumor growth (Fig 7C) and also reduced the expression of mDia2 in the stroma (Fig 7D).

A strong tumor-suppressive effect of activin A-mDia2 inhibition was also seen when tumors formed by SCC13 Act cells were treated *ex vivo* with recombinant human FST and/or SMIFH2, an mDia2 inhibitor (Isogai *et al*, 2015b). *Vice versa*, recombinant activin A further promoted cell proliferation in tumor explants (Fig 7E). In tumor spheroid models where SCC13 Act cells with fluorescence labeled membranes were cultured with primary human fibroblasts, FST or SMIFH2 suppressed spheroid expansion (Fig 7F).

Finally, the therapeutic potential of activin A inhibition was shown *in vivo*, since injection of FST into A431 ear xenograft tumors strongly reduced their growth. This correlated with downregulation of mDia2 expression in the stroma (Fig 7G). Together, these findings reveal a crucial role of the activin A-mDia2 signaling pathway in squamous carcinogenesis by reprogramming of normal fibroblasts into CAFs. Therefore, inhibition of this axis is a promising strategy for the treatment of squamous cancers and other activin A-dependent malignancies (Fig 7H).

## Discussion

Here, we show that activin A captures major players in normal cell biology to reprogram fibroblasts into pro-tumorigenic CAFs. Therefore, blocking of this signaling axis offers exciting opportunities for cancer prevention and treatment. Importantly, the activin signaling pathway discovered in this study was not activated by TGF-β1,

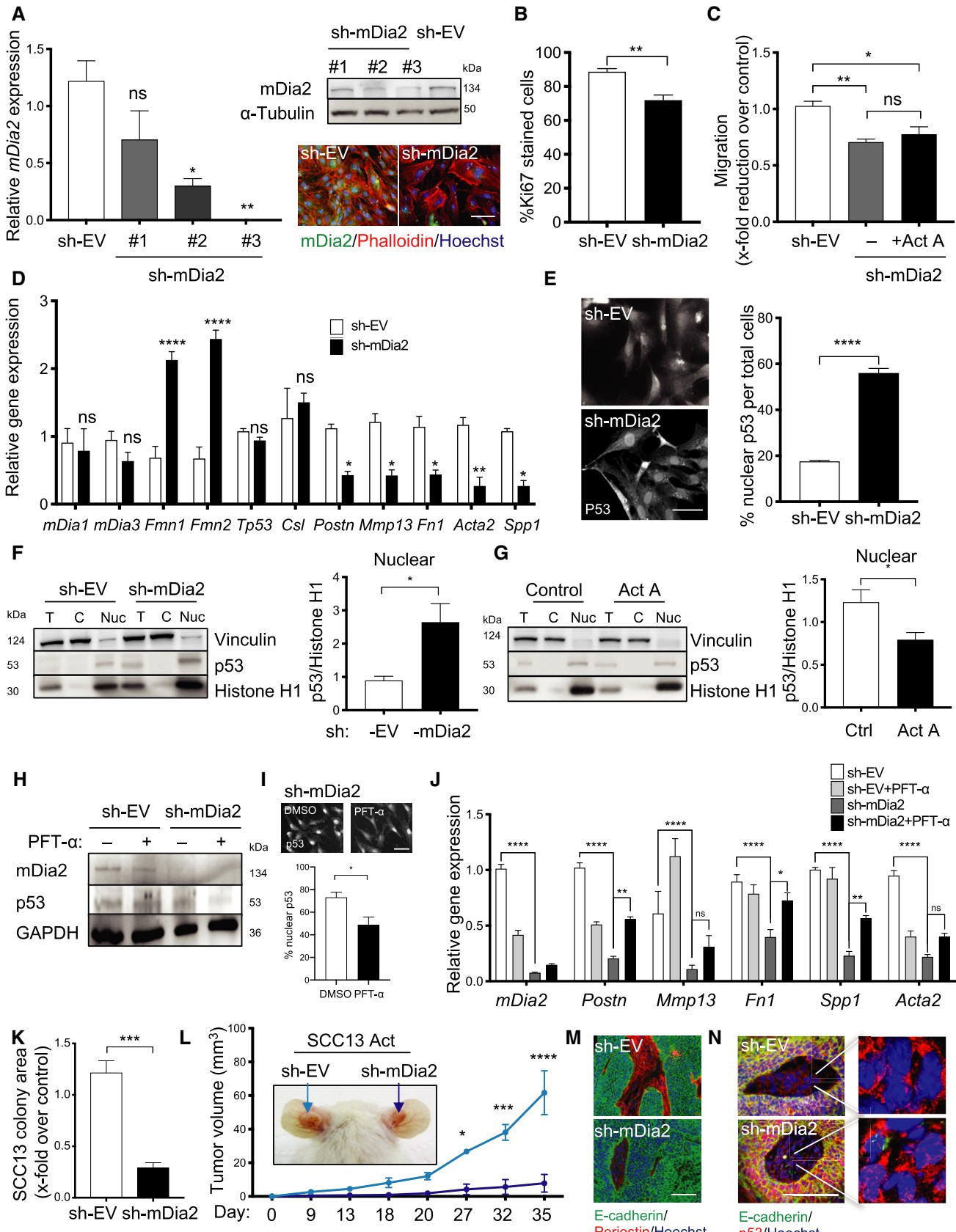

**Figure 6.**

**Figure 6. Knock-down of mDia2 in fibroblasts suppresses squamous cell malignancy and tumor growth via p53.**

A    Immortalized mouse fibroblasts were transduced with a lentiviral vector expressing different mDia2-targeting shRNAs (shmDia2 #1-3) or empty vector (sh-EV). The knock-down was confirmed by qRT–PCR (N = 3), Western blot, and immunofluorescence.

B, C  Ki67 staining (B) and migration in scratch assays (C) of mDia2 knock-down and control mouse fibroblasts. N = 3.

D    qRT–PCR for *mDia1*, *mDia3*, *Fmn1*, and *Fmn2* and several CAF genes relative to *Rps29* using RNA from control (sh-EV) and mDia2 knock-down (sh-mDia2) fibroblasts. N = 3.

E    Representative images of mDia2 knock-down and control mouse fibroblasts stained for p53 (left), and quantification of cells showing accumulation of p53 in the nucleus (right). N = 3.

F, G  Representative Western blots for p53, vinculin, and histone H1 using nuclear and cytoplasmic fractions and total lysate of mDia2 knock-down (F) or activin A-treated fibroblasts (6 h, 20 ng/ml) (G) and control mouse fibroblasts and quantification of the nuclear p53/histone H1 ratio. Vinculin and histone H1 were used as loading controls for the cytoplasmic and nuclear fractions, respectively. N = 6.

H    Representative Western blots for p53, mDia2, and GAPDH using total lysates of mDia2 knock-down and control mouse fibroblasts treated for 24 h with pifithrin-α (PFT-α) (15 μM).

I    Representative p53 immunofluorescence images of mDia2 knock-down mouse fibroblasts treated with PFT-α or vehicle (DMSO) (top) and quantification of cells showing accumulation of p53 in the nucleus (bottom). N = 3.

J    qRT–PCR for *mDia2*, *Postn*, *Mmp13*, *Fn1*, *Spp1*, and *Acta2* relative to *Rps29* using RNA from mDia2 knock-down and control fibroblasts treated with PFT-α or vehicle. N = 3.

K    Colony size of SCC13 Act cells (clone 2) after co-culture with either immortalized mDia2 knock-down (sh-mDia2) or control (sh-EV) mouse fibroblasts. N = 4.

L    Representative pictures of 5-week-old tumors formed upon intradermal co-injection of SCC13 Act cells (clone 2) with sh-EV or sh-mDia2 knock-down immortalized mouse fibroblasts. Graph shows tumor volume at different time points of tumor development. N = 3–5 tumors per group.

M, N  Representative immunofluorescence images of sections from ear skin tumors formed by SCC13 Act (clone 2) co-injected with sh-EV or sh-mDia2 fibroblasts, stained for E-cadherin (green) and periostin or p53 (red), and counterstained with Hoechst (blue).

Data information: Bar graphs show mean ± SEM. *P < 0.05, **P < 0.01, ***P < 0.001, ****P < 0.0001, unpaired Student's *t*-test (B, E–G, I, and K) and one-way ANOVA with Bonferroni post hoc test (A, C), two-way ANOVA with Bonferroni post-test (D, J, L). Scale bars: 100 μm. Exact *P*-values are provided in Dataset EV3.
Source data are available online for this figure.

another key player in CAF formation (Calon *et al*, 2014). Rather, TGF-β1 promotes myofibroblast differentiation, strongly suggesting that activin A and TGF-β1 induce distinct types of CAFs.

A few additional drivers of the CAF phenotype have been described, e.g., loss of Notch/CSL signaling, in particular when combined with p53 downregulation (Procopio *et al*, 2015). Furthermore, DNA damage or adrenergic signaling in tumor cells induced CAF activation in mammary and ovarian cancer models, which was at least in part mediated by *Inhba* expression in the tumor cells (Fordyce *et al*, 2012; Nagaraja *et al*, 2017). These results indicate important connections between different CAF inducers.

Through transcriptional activation of *mDia2*, activin directly promotes the migratory capacity of fibroblasts and also induces the expression of CAF markers, which include various cytokines and ECM components. These activities are likely to be beneficial during wound healing, when activin A is transiently upregulated and promotes the healing process (Munz *et al*, 1999). However, upon long-term activation and in the presence of mutated cancer cells, the secretome and the matrisome of activin A-exposed fibroblasts promoted malignant features of SCC cells in an mDia2-dependent manner. This is consistent with the finding that high *mDia2* expression supported ECM remodeling and invasion of mammary CAFs in culture (Calvo *et al*, 2013). Remarkably, overexpression of mDia2 alone induced CAF properties in normal fibroblasts. Thus, mDia2 is required for various pro-tumorigenic activities of activin A and sufficient for establishment of a CAF phenotype. The *in vivo* relevance of these findings was demonstrated in xenograft models, where co-injection of mDia2 knock-down fibroblasts with SCC cells overexpressing activin A strongly diminished the potent effect of activin A on tumor growth and progression. The strong correlation between increased activin A-mDia2 expression and reduced survival of patients with different malignancies supports the clinical relevance of our findings.

The observations that mDia2 binds p53 and that silencing of mDia2 in fibroblasts resulted in p53 accumulation in the nucleus and

concomitant lower expression of CAF markers shed light on the mechanism whereby activin A regulates the CAF phenotype. p53 exerts a non-autonomous tumor-suppressive role in fibroblasts through suppression of paracrine-acting factors that promote tumor formation (Moskovits *et al*, 2006; Addadi *et al*, 2010). Accordingly, p53 downregulation in fibroblasts promoted prostate and skin tumor growth (Addadi *et al*, 2010; Procopio *et al*, 2015). Interestingly, lung cancer cells were shown to secrete unknown factors that suppress p53 activity in fibroblasts. Our results suggest that activin A may be one of these factors, since it reduced the levels of p53 in the nucleus. They further show that induction of pro-tumorigenic, p53-repressed CAF genes in fibroblasts by activin A is dependent on *mDia2* and that mDia2 overexpression is sufficient to promote expression of these genes. The functional relevance of this interaction for the *in vivo* situation is supported by the negative correlation of activin A overexpression in mouse skin with the expression of p53 target genes.

In recent years, non-invasive treatments of NMSC became available, including topical application of the TLR7 agonist imiquimod, and systemic use of immune check point inhibitors, which rely on immune cell-mediated apoptosis of cancer cells (Love *et al*, 2009; Migden *et al*, 2018), or of hedgehog signaling inhibitors for BCC, which directly target the tumor cells (Gutzmer & Solomon, 2019). Our data suggest targeting of the fibroblast-cancer cell cross-talk as a promising alternative. This would offer clinical advantages, since fibroblasts are genetically more stable and less resistant to therapeutic intervention compared with tumor cells (Pure & Lo, 2016). Both follistatin and SMIFH2 efficiently blocked the pro-tumorigenic effect of activin A through inhibition of the activin A-mDia2 axis. Consistent with these findings, follistatin suppressed metastasis in mouse breast and lung cancer models (Ogino *et al*, 2008; Seachrist *et al*, 2017) and improved the efficacy of platinum chemotherapy in the management of activin A-induced lung adenocarcinoma (Marini *et al*, 2018). However, a role of fibroblasts in the therapeutic effect of activin A inhibition has as yet not been reported and relevant

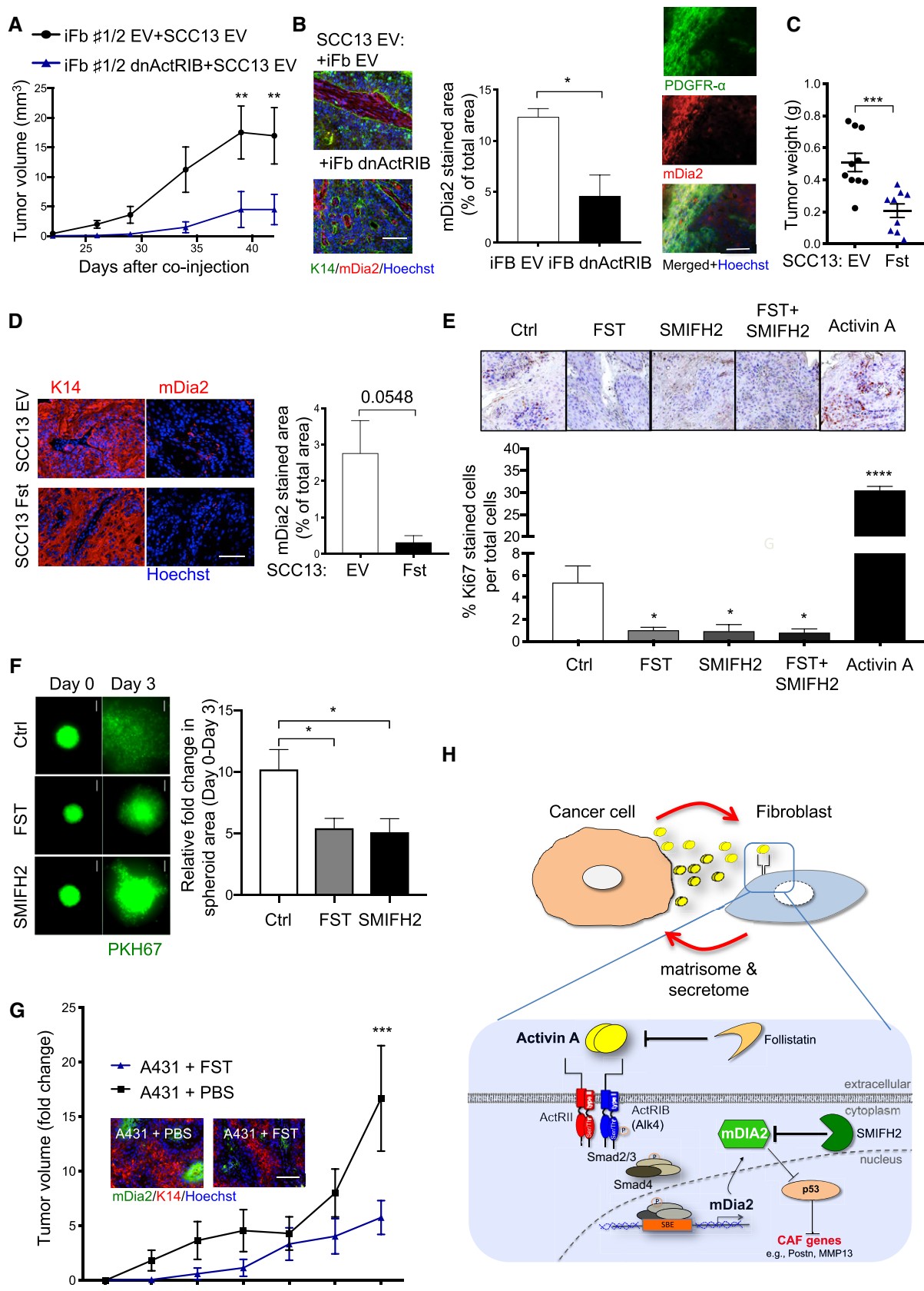

Figure 7.

**Figure 7.** **Inhibition of activin A-mDia2 signaling suppresses squamous tumor growth.**

A Tumor volume at different time points after co-injection of SCC13 cells with EV or dnActRIB fibroblasts. *N* = 5 tumors.

B Representative immunofluorescence images of sections from ear skin tumors formed by SCC13 cells co-injected with EV or dnActRIB fibroblasts, stained for K14 (left) or PDGFRα (right) (green) and mDia2 (red), and counterstained with Hoechst (blue). Bar graph shows the percentage of mDia2-positive area. *N* = 3.

C SCC13 EV or SCC13 Fst cells were injected into the ear skin of NOD/SCID mice, and tumor weight was determined 12 weeks after injection. *N* = 9–10 tumors.

D Representative immunofluorescence images of sections from ear skin tumors formed by SCC13 EV or SSC13 Fst cells, stained for K14 or mDia2 (red), and counterstained with Hoechst (blue). Bar graph shows the percentage of mDia2-positive area. *N* = 3.

E Tumors formed by SCC13 Act cells were treated *ex vivo* for 6 h with follistatin (FST) (50 ng/ml) and/or SMIFH2 (10 μM), or with activin A (20 ng/ml) and stained for Ki67. *N* = 3 tumor explants, *n* = 4 histological sections.

F Representative images of spheroids formed by SCC13 Act cells labeled with PKH67 (green) with primary human fibroblasts embedded in collagen gels and graph showing increase in the spheroid area between day 0 and day 3. Spheroids were treated with either FST (50 ng/ml) or SMIFH2 (10 μM) or left untreated. *N* = 6–8 tumor spheroids.

G Increase in A431 tumor volume ($mm^3$) compared to day 0 (time when the tumors were first injected with FST, approximately 3–4 weeks after injection of the cells) upon treatment of the tumors with FST (100 ng per injection) or vehicle (PBS). *N* = 3–5 tumors per group. Representative immunofluorescence images of the FST or vehicle-treated tumors, stained for mDia2 (green) and K14 (red), counterstained with Hoechst (blue).

H Proposed role of activin A secreted by cancer cells in inducing a pro-tumorigenic CAF phenotype in fibroblasts. Activin A signaling (blocked by follistatin or dnActRIB (not shown in the figure) induces upregulation of mDia2 (inhibited by SMIFH2), which in turn contributes to tumor development by reducing the nuclear p53 pool and promoting the expression of CAF marker genes, fibroblast proliferation, and migration.

Data information: Bar graphs show mean ± SEM. ns *P* > 0.05, *\*P* < 0.05, *\*\*P* < 0.01, *\*\*\*P* < 0.001, *\*\*\*\*P* < 0.0001 (unpaired Student's *t*-test (B-D), one-way ANOVA with Bonferroni post-test (E, F), two-way ANOVA with Bonferroni post-test (A, G)). Scale bars: 100 μm. Exact *P*-values are provided in Dataset EV3.

target genes remain to be identified. Blocking of mDia2 in fibroblasts may well have a similar effect, but prolonged exposure to SMIFH2 leads to p53 downregulation (Isogai *et al*, 2015b), and reports showing that downregulation of mDia2 in certain cancer cells promotes their motility have to be considered and may limit the usefulness of mDia2 inhibitors (Di Vizio *et al*, 2009; Creekmore *et al*, 2011; Pettee *et al*, 2014). Therefore, inhibition of activin A or its receptors is the more promising approach. In some cancers, such as melanoma, inhibition of activin A additionally promotes antitumor immunity, e.g., via its action on natural killer cells (Donovan *et al*, 2017; Rautela *et al*, 2019). Interestingly, soluble activin type II receptors are in clinical trials for the treatment of activin A-induced cancer cachexia or bone destruction, and revealed low toxicity (Pearsall *et al*, 2008; Zhou *et al*, 2010). These and our results raise the exciting possibility of developing novel targeted therapies that combine effective inhibition of tumor growth with minimal cancer-associated morbidity.

## Materials and Methods

### Animals

NOD/SCID (NOD.CB17-Prkdc^scid/NCrCrl) mice, PDGFRα-eGFP transgenic mice (Hamilton *et al*, 2003), and transgenic mice overexpressing *INHBA* in keratinocytes (Act mice) (Munz *et al*, 1999) with or without the HPV8 oncogenes (Schaper *et al*, 2005; Antsiferova *et al*, 2017) were used. They were housed under specific pathogen-free conditions and received food and water *ad libitum*. Mouse maintenance and all animal experiments had been approved by the veterinary authorities of Zurich, Switzerland.

### Primary cells and cell lines

SCC13 cells (human cutaneous squamous cell carcinoma cell line; Rheinwald & Beckett, 1981) were kindly provided by Dr. Petra Boukamp, Leibniz Institute for Environmental Research, Düsseldorf, Germany, and verified to be pure SCC13 cells by the Deutsche Sammlung von Mikroorganismen und Zellkulturen (DMSZ) GmbH, Braunschweig, Germany.

A431 cells (epidermoid vulvar carcinoma cells, Giard *et al*, 1973) were purchased from Sigma, Munich, Germany.

Primary foreskin fibroblasts were established from human foreskin and kindly provided by Dr. Hans-Dietmar Beer, University of Zurich, Switzerland.

All mouse primary cells and immortalized fibroblast lines were established in our laboratory as previously described (Hiebert *et al*, 2018).

All cells were cultured in Dulbecco's modified Eagle's medium (DMEM) supplemented with 10% fetal calf serum (FCS) and penicillin/streptomycin.

### 3D organotypic skin cultures

Scaffold-free organotypic 3D cultures were established according to Berning *et al* (2015). For this purpose, $1.86 \times 10^6$ human fibroblasts were seeded onto 6-well inserts with a 0.4 μm transparent polyethylene terephthalate (PET) membrane in 1.5 ml of sterile-filtered Medium A (DMEM with high glucose/L-glutamine and DMEM/F-12 GlutaMAX™ in a 1:1 ratio), supplemented with 0.2 mg/ml 2-phospho-L-ascorbic acid, 1 ng/ml TGF-β1, 2.5 ng/ml epidermal growth factor (Sigma-Aldrich), 5 ng/ml fibroblast growth factor 2 (PeproTech), and 5 μg/ml insulin (Sigma-Aldrich). Outside of the insert, another 12 ml of Medium A was added to Falcon® 6-well Deep Well tissue culture-treated polystyrene plates (Corning). Seeding of fibroblasts was repeated after 2 and 4 days. Medium A inside and outside the 6-well insert was replaced every second day until day 31. On day 32, $0.928 \times 10^6$ SCC13 or A431 cells were seeded on top of the dermal layer with 1.5 ml sterile-filtered Medium B [DMEM with high glucose/L-glutamine and DMEM-F12 GlutaMAX™ in a 1:1 ratio, supplemented with 0.2 mg/ml 2-phospho-L-ascorbic acid, 0.0084 μg/ml cholera toxin, 0.4 μg/ml hydrocortisone, with our without 1 μg/ml doxycycline (all from Sigma-Aldrich)] according to the experimental setup. In addition, another 12 ml of Medium B was added to the well. After 3 days, the medium was removed from the insert, and the outside medium was replaced by 12 ml of

fresh medium B. Cultures on the inserts were then cultured at the air–liquid interface. The outside medium was replaced every second day by 12 ml of Medium B until collection (day 60). Mature cultures were harvested, bisected, and embedded in tissue freezing medium (Leica Microsystems) for further analysis by histochemical or immunofluorescence staining.

### Cell proliferation assay

Cell proliferation was determined by counting of cells or by analysis of BrdU incorporation (Hiebert et al, 2018).

### Cloning of cDNAs into the pInducer20 plasmid

The coding sequences for INHBA, FST, or dnActRIB were amplified from plasmids using the primers listed in Appendix Table S1, cloned into the pENTR/D-TOPO vector (K240020, Thermo Fisher Scientific), and subsequently into the pInducer20 vector (kindly provided by Dr. S. Elledge) using the LR Clonase II Enzyme mix (#11791019, Thermo Fisher Scientific).

### Lentiviral production

HEK293T cells at 30% confluency were transfected overnight in DMEM/10% FBS without P/S using jetPEI Transfection Reagent (#101-40N, Polyplus-Transfection SA). For each 10 cm dish, 5 µg DNA was incubated with 1.75 µg pMD2.G and 3.25 µg pCMV-dR8.91 (#12259 Addgene). On the next day, the medium was replaced by DMEM/10% FBS/P/S. Cells were cultured for at least 48 h to allow production of the virus. The supernatant was filtered through a Filtropur S 0.2-µm filter (#83.1826.001, Sarstedt) and stored at −80°C.

### Lentiviral transduction and generation of stably transfected cell cultures

Fibroblasts, SCC13, or A431 cells were grown to 60–70% confluency in 6 cm dishes and incubated with 1 ml/dish of viral supernatant in DMEM/10%FBS/P/S supplemented with 8 µg/ml of polybrene for 6 h at 37°C/5% $CO_2$. Afterward, fresh medium with polybrene was added. Two days later, the selection process was started by addition of 1 µg/ml G418/geneticin (#11811031, Thermo Fisher Scientific) for pInducer20-transduced cells or 0.5 µg/ml puromycin (P8833, Sigma-Aldrich) for pLKO.1-transduced cells.

### Lentivirus-mediated expression of mDia2 shRNAs

Immortalized mouse dermal fibroblasts were infected with lentiviral shRNA vectors designed by The RNAi Consortium (TRC). Three shRNAs directed against mouse mDia2 in the pLKO.1 lentiviral vector were tested (Appendix Table S2). Primary human dermal fibroblasts were infected with lentiviruses pLKO.1-shmDia2 #1; TRCN0000150903, which target both *mDia2* and *DIAPH3* (Isogai et al, 2015a,b).

### Transient transfection of fibroblasts

Wild-type Flag-tagged mDia2 cDNA (Isogai et al, 2015a, 2016) was subcloned into the retroviral pMX vector (Cell Biolabs). One microgram pMX Flag-mDia2 or empty vector and Lipofectamine™ 2000 (#11668030, Invitrogen) were incubated with Opti-MEM™ I Reduced Serum Medium (#31985062, Thermo Fisher Scientific) at room temperature for 20 min. The mixture was then added to primary human fibroblasts. After 6-h incubation, medium was replaced by normal culture medium containing 10% FBS. Cells were allowed to recover for 48 h prior to analysis.

### Collagen gel contraction assay

Gel contraction was performed in 24-well plates coated with 1% BSA at 37°C for 1 h. For each gel, 120 µl HEPES, 26 µl 10× DMEM, 16 µl Milli-Q water, and 240 µl of cell suspension (420,000 cells/ml) were mixed. 198 µl of TeloCol Type I Bovine Collagen Solution (#5026, Advanced BioMatrix) was added to the well. The plate was incubated at 37°C and 5% $CO_2$ for 90 min to solidify. Afterward, 500 µl of DMEM with 1% FBS and 1% P/S were added to the gels. Gels were incubated with activin A (20 ng/ml) and/or TGF-β1 (1 ng/ml; #100-21, PeproTech) at 37°C/5% $CO_2$, and contraction was monitored for 12–72 h.

### Culture of SCC13 cells with fibroblasts

Equal numbers of SCC13 cells and fibroblasts were seeded together on glass slides in 24-well plates. The cells were co-cultured in complete DMEM with DOX (1 µg/ml) for 7 days and subsequently stained with K14 antibody to measure the SCC13 colony number/area.

### Spheroid formation and growth assay

Twenty microlitre growth medium containing 2,000 SCC13 Act cells, which had been pre-treated with PKH67 (Sigma-Aldrich) for cell membrane labeling, was placed on the lids of 6-cm culture plates. To prevent dehydration, 5 ml PBS was added to the bottom. Cells were incubated at 37°C and 5% $CO_2$ for 72 h. Upon aggregate formation in the drops, spheroids were collected and transferred onto a collagen gel pre-seeded with primary human fibroblasts. They were incubated for 3 additional days in complete DMEM with DOX. Fluorescence images were captured on a ZOE™ Fluorescent Cell Imager (Bio-Rad), and spheroid outgrowth was quantified using ImageJ software. To analyze anchorage-independent growth, spheroids were incubated in different culture media and analyzed for increase in spheroid area.

### RNA isolation and qRT–PCR

RNA isolation and quantitative qRT–PCR were performed as described (Hiebert et al, 2018) using primers listed in Appendix Table S3. Values obtained for the first control were set to 1.

### Preparation of protein samples and Western blot analysis

Preparation of protein lysates and Western blot were performed using standard procedures. Proteins from conditioned medium were precipitated by addition of trichloroacetic (10% v/v), pelleted by centrifugation at 20,000 g for 15 min at 4°C, washed twice with

ice-cold acetone, air-dried, and resuspended in 25 µl 2× Laemmli buffer.

Antibodies against INHBA (sc166503, Santa Cruz, 1:500 diluted), mDia2 (recognizing both mDia2 and DIAPH3; 1:5,000 diluted) (Isogai *et al*, 2015a,b), αSMA (A2547, Sigma-Aldrich, 1:500 diluted), p53 (sc-126, Santa Cruz or AB17990, Abcam, 1:500 diluted), MYC-tag (AB9106, Abcam, 1:1,000 diluted), α-tubulin (T5168, Sigma-Aldrich. 1:10,000 diluted), vinculin (V4505, Sigma-Aldrich, 1:2,000 diluted), histone H1 (AB134914, Abcam, 1:500 diluted), and GAPDH (#5G4, HyTest, 1:10,000 diluted) were used. Secondary antibody was anti-rabbit or anti-mouse IgG (W4011 and W4021, Promega, 1:8,000 diluted) conjugated with horseradish peroxidase, and chemiluminescence was determined using the WesternBright ECL Detection System (Advansta). Bands were visualized using Fusion Solo 6S (Witeg AG), and intensity was quantified with ImageJ software (National Institutes of Health).

## Immunoprecipitation

Protein lysates from $1 \times 10^7$ primary human foreskin fibroblasts were prepared using 0.5 ml cell lysis buffer (0.1% NP40 in PBS with 1× protease inhibitor cocktail (Roche)) and collected by centrifugation at 14,000× *g* for 15 min at 4°C. 200 µl of protein lysate was incubated with 1–2 µg of normal rabbit IgG (#2729, Cell Signaling, 1:1,000 diluted) or DIAPH3 antibody (Isogai *et al*, 2015a,b; 1:1,000 diluted) under rotation overnight at 4°C. The immunocomplex was then mixed with 20 µl solution containing Dynabeads™ Protein A (#10002D, Thermo Fisher Scientific). Antibody-bound beads were incubated for 20 min at room temperature on a rotator, recovered using a magnetic separation rack, and washed five times with 0.5 ml lysis buffer. Immunoprecipitated proteins were eluted with 2× Laemmli sample buffer.

## Gelatin zymography

Conditioned media were collected from the 3D organotypic skin cultures, centrifuged, and concentrated using an Amicon Ultra-4 Centrifugal Filter Unit device with 3 kDa cut-off (UFC800308, Merck) at 4,500 *g* for 30 min at 4°C. Supernatants were subjected to electrophoresis in a 7.5% SDS–polyacrylamide gel co-polymerized with gelatin (1 mg/ml, Sigma-Aldrich). Enzymatically active MMPs were detected as transparent bands in the Coomassie Brilliant Blue-stained gelatin gel.

## Nucleus/cytosol fractionation

Cells were lysed with 350 µl 0.1% NP40-PBS with 1× protease inhibitor cocktail (#11697498001, Roche) on ice, and lysates were centrifuged at 17,000 *g* for 3 min. The supernatant included cytoplasmic extract. The nuclear pellets were washed with 800 µl lysis buffer, centrifuged at 17,000 *g* for 3 min (3× each), and resuspended in 70 µl of 2× Laemmli buffer.

## Histology and immunostaining

Histological analysis and immunostainings were performed as described (Hiebert *et al*, 2018) using the antibodies listed in Appendix Table S4.

## Migration assays

Chemotactic transwell migration was assessed as described (Antsiferova *et al*, 2017). Primary mouse fibroblasts were incubated for 24 h in DMEM/1% FBS for baseline measurements and in DMEM/1% FBS supplemented with activin A (#338-AC, 1–20 ng/ml) (R&D Systems) in the lower compartment of the chamber.

For transwell migration assays, SCC13 cells were seeded on the insert and fibroblasts were seeded on the lower chamber. Migration of cells toward the factors secreted by fibroblasts was determined.

To determine the capacity of the matrix deposited by fibroblasts to stimulate SCC cell migration, SCC13 cells were seeded on culture inserts (#80209, Ibidi) that were placed on top of the matrix that had been deposited by fibroblasts. The insert was then removed for migration assays.

For scratch assays, cells were grown to 100% confluency and treated with 2 µg/ml mitomycin C for 2 h. One or several scratches were made into the cell layer using a sterile 200 µl pipette tip. Dead cells and debris were washed off with pre-warmed PBS. The same area was photographed directly after scratching and at different time points thereafter.

## FACS and RNA sequencing

Cell dissociation, FACS, RNA sequencing, and bioinformatics analysis were performed as previously described (Antsiferova *et al*, 2017). Briefly, fibroblasts from the ear skin of female mice (13–15 weeks old, CD1 genetic background) were isolated based on the fibroblast marker CD140a (antibody #135908, BioLegend). RNA was isolated using the RNeasy Micro Kit (Qiagen), analyzed via TapeStation (Agilent Technologies), and samples with RIN > 7 were adjusted to 100 ng and subjected to RNA sequencing via poly-A enrichment, True-Seq library preparation, and sequencing on an Illumina HiSeq 2500 instrument (Illumina). Quality control via FastQC showed rRNA mapping rates of < 5% and over 30 million reads for all samples. RSEM-normalized gene count data were analyzed for genes that are regulated by activin A in the presence or absence of the HPV8 transgene ($n = 3$).

## GEO2R and KMPlotter (Kaplan–Meier Plotter) analysis

Using the GEO2R analysis tool in the GEO database, *mDia2/DIAPH3* and *INHBA* gene expression values were acquired from public mRNA datasets of human cancers (Barrett *et al*, 2013).

Survival graphs were generated using the KMPlotter gene expression-based survival analysis web application by integrating gene expression and clinical data simultaneously with statistical analysis of log-rank *P* value (Nagy *et al*, 2018).

## Preparation of fibroblast secretomes and matrisomes

Cells were plated at 70–90% confluency in DMEM/10% FBS/P/S. On the following day, they were pre-treated with 2 µg/ml mitomycin C and cultured in starvation medium (DMEM/1%FBS/P/S) with 1 µg/ml DOX for an additional 3 days. Conditioned media and ECM were prepared as described (Hiebert *et al*, 2018).

                                                        

## Skin tumorigenesis assays

Tumorigenesis assays in mouse ear skin were performed as described (Procopio *et al*, 2015). SCC13 or A431 cells overexpressing *INHBA* or *FST* or control cells were treated with DOX for 24 h. In brief, 3 μl cell suspension of $2 \times 10^5$ cells in Hank's buffer was injected intradermally into the ear of male NOD/SCID mice, 10–15 weeks of age. For tumor cell/fibroblast co-injection experiments, we injected $10^5$ cancer cells and an equal number of fibroblasts. Mice were injected with DOX (20 mg/kg) every other day. Tumor formation was monitored over 5 weeks.

For follistatin treatment experiments, A431 cells were injected intradermally and tumor formation was observed during 4 weeks. Tumors were then treated with follistatin (100 ng) or vehicle (PBS) every 3–4 days by direct injection into the tumor. Four days after the last injection, tumors were harvested.

## Human skin biopsies

Normal human skin and skin cancer samples were obtained anonymously from the Department of Dermatology, University Hospital of Zurich (in the context of the Biobank project), approved by the local and cantonal Research Ethics Committees. Normal skin was from healthy adult volunteers or from the edges of skin tumors. BCC or SCC was diagnosed by experienced pathologists. Informed consent was obtained from all subjects and the experiments conformed to the principles set out in the WMA Declaration of Helsinki and the Department of Health and Human Services Belmont Report.

## Prediction of SMAD2/3-mediated transcriptional regulation of *mDia* genes

The numbers of predicted SMAD2/3 binding sites 100 kb upstream of the TSS or within the genomic region of the *DIAPH1-3* genes were determined using the GTRD ChIP-seq database (http://gtrd.biouml.org/) (Yevshin *et al*, 2017).

## Chromatin immunoprecipitation (ChIP)

Primary mouse dermal fibroblasts (approx. $5 \times 10^8$ cells) at 80–90% confluence, which had been pre-treated with activin A (20 ng/ml) or TGF-β1 (1 ng/ml) for 6 h, were collected, resuspended in DMEM, and subjected to ChIP as previously described (Wilanowski *et al*, 2008). The purified DNA fragments were used for PCR amplification using the primers listed in Appendix Table S5.

## *Ex vivo* tumor explant cultures

Ear tumors were cut into fragments of approximately 0.5 cm$^2$ and cultured at the air–liquid interface in DMEM/10% FBS supplemented with 5 μg/ml insulin (I5500), 0.1 nM cholera toxin (C8052), 10 ng/ml epidermal growth factor (E4127), 50 IU/ml P/S, and 0.4 μg/ml hydrocortisone (#386698) (all from Sigma-Aldrich). After overnight incubation, they were treated with activin A (20 ng/ml), follistatin (50 ng/ml) (#120-13, PeproTech), SMIFH2 (10 μM) (S4826, Sigma-Aldrich), or combinations for 6 h and embedded in paraffin. Sections were analyzed by immunohistochemistry using a Ki67 antibody (ab15580, Abcam, 1:500 dilution).

### The paper explained

#### Problem

Skin cancer is by far the most common type of cancer worldwide. Since stromal alterations strongly contribute to the high incidence of skin cancer, treatments targeting the cancer microenvironment appear very attractive. In this context, the role of cancer-associated fibroblasts (CAFs) is being increasingly recognized. However, the molecular mechanisms of CAFs in skin tumorigenesis are still poorly characterized.

#### Results

We discover that activin A promotes skin tumorigenesis via reprogramming of fibroblasts into CAFs. This effect is mediated to a large extent via activin A-induced upregulation of the formin mDia2. Activation of the activin A-mDia2 axis negatively correlates with survival in different human cancers and blocking of this axis at multiple levels inhibited skin carcinogenesis in mouse models.

#### Impact

Our study suggests inhibition of the activin A-mDia2 axis at multiple levels as a therapeutic strategy in cancer patients. Since activin receptor inhibitors are in clinical trials for the treatment of activin A-induced cancer cachexia and muscle wasting, our findings raise the exciting possibility that blocking of activin A may inhibit both cancer growth and cancer-induced morbidity.

## Ingenuity pathway analysis

Lists of genes differentially expressed in FACS-isolated fibroblasts from activin-overexpressing versus control mice were uploaded to Ingenuity® Pathway Analysis (Qiagen). Core analysis was performed with filtered lists of significantly up- and downregulated genes with |Log$_2$Ratio|>1 and FDR < 0.05 in three comparisons: Act/wt versus wt/wt, Act/HPV versus wt/HPV, and Act/HPV versus wt/wt. Data tables associated with "Diseases & Functions" and "Upstream Regulators" were exported to identify highly activated enriched functions and regulators in activin A-exposed fibroblasts. Selected functions were arranged in a spreadsheet, and their activation *z*-scores were color-coded for visualization. Original IPA output data with unfiltered "Diseases & Functions" and "Upstream Regulators" for all comparisons are provided in Dataset EV1. Additional enrichment of upregulated genes was done using Reactome and matrisome databases (Naba *et al*, 2012).

## Gene set enrichment analysis (GSEA)

Sets of significantly up- and downregulated genes were generated by analyzing published microarray data of CAFs or total stroma of various types of human cancer (see Dataset EV2). Original gene sets were uploaded to GSEA and filtered to those mapped by gene symbol and present in the tested datasets. The gene sets were tested against the GSEA-generated ranked gene lists of the individual genotype comparisons Act/wt versus wt/wt, Act/HPV versus wt/HPV, Act/HPV versus wt/wt, and the combined [all]/Act/[all] versus wt/[all] comparison. Four separate GSEA runs were completed: (i) testing all individual genotype comparisons against gene sets and (ii) testing the [all]/Act versus [all]/wt comparison against gene sets. GSEA results were organized in tables, and the normalized enrichment scores (NES) and false discovery rate (FDR) values were color-coded for visualization.

Original and filtered gene sets, ranked gene lists, and original GSEA output data for all experiments are provided in Dataset EV2.

### Statistics

Statistical analysis was performed using the Prism software, version 7 for Mac OS X or Windows (GraphPad Software Inc). For comparison of two groups, unpaired Student's *t*-test was performed; for comparison of more than two groups, one-way or two-way ANOVA and Bonferroni's multiple comparisons test were used. ns $P > 0.05$, $*P < 0.05$, $**P < 0.01$, $***P < 0.001$ $****P < 0.0001$.

## Data availability

The RNA-seq data have been deposited in NCBI's Gene Expression Omnibus (GEO; https://www.ncbi.nlm.nih.gov/geo/) (GSE130017).

**Expanded View** for this article is available online.

## Acknowledgements

We thank Drs. Pino Bordignon and Gian-Paolo Dotto, University of Lausanne, for help with the ear tumorigenesis assays; Dr. Petra Boukamp, Leibniz Institute for Environmental Medicine, Düsseldorf, for help with the 3D organotypic cultures and for providing SCC13 cells; Catharine Aquino Fournier and Lennart Opitz from the Functional Genomics Center Zurich for RNA sequencing; Dr. Hans-Dietmar Beer, University of Zurich, for primary human skin fibroblasts; Dr. Steve Elledge, Harvard University Medical School, Boston, for the pInducer20 vector; and Dr. Stephen Jane, Monash University, Melbourne, Australia, and Dr. Richard Grose, Barts Cancer Institute, London, UK, for helpful suggestions on the manuscript. This work was supported by grants from Cancer Research Switzerland (KFS-4510-08-2018 to S.W.), the Swiss National Science Foundation (31003A_169204 to S.W.), and University Medicine Zurich (Project SKINTEGRITY to S.W.).

## Author contributions

MC and SW designed the study; MC, MW, NM, RO, LF, DA-N, and MA performed experiments and analyzed data; RD provided the human skin cancer samples and clinical skin cancer expertise. MI contributed to the design of the mDia2 experiments and provided mDia2 expertise and reagents; SW and MC wrote the manuscript; and SW acquired the funding. All co-authors made important comments on the manuscript.

## Conflict of interest

The authors declare that they have no conflict of interest.

## For more information

(i) Website SKINTEGRITY Collaborative Skin Research Program: https://www.hochschulmedizin.uzh.ch/en/projekte/skintegrity.html

(ii) Website Werner lab, ETH Zurich: https://mhs.biol.ethz.ch/research/werner/research-werner.html

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
