## [Review Process File · EMBO Molecular Medicine]

A paracrine activin A–mDia2 axis promotes squamous carcinogenesis via fibroblast reprogramming

Michael Cangkrama, Mateusz Wietecha, Nicolas Mathis, Rin Okumura, Luca Ferrarese, Dunja Al-Nuaimi, Maria Antsiferova, Reinhard Dummer, Metello Innocenti, and Sabine Werner

Review timeline:

Submission date:	20th Sep 2019
Editorial Decision:	17th Oct 2019
Revision received:	24th Dec 2019
Editorial Decision:	30th Jan 2020
Revision received:	6th Feb 2020
Accepted:	10th Feb 2020

Editor: Lise Roth

Transaction Report:

1st Editorial Decision

17th Oct 2019

Thank you for the submission of your manuscript to EMBO Molecular Medicine. We have now received feedback from the three reviewers who agreed to evaluate your manuscript. As you will see from the reports below, the referees acknowledge the interest of the study and are overall supporting publication of your work pending appropriate revisions.

Addressing the reviewers' concerns in full will be necessary for further considering the manuscript in our journal, and acceptance of the manuscript will entail a second round of review. EMBO Molecular Medicine encourages a single round of revision only and therefore, acceptance or rejection of the manuscript will depend on the completeness of your responses included in the next, final version of the manuscript. For this reason, and to save you from any frustrations in the end, I would strongly advise against returning an incomplete revision.

***** Reviewer's comments *****

Referee #1 (Comments on Novelty/Model System for Author):

Very thorough study combining mouse in vivo studies and in vitro studies with murine and human cells.

Referee #1 (Remarks for Author):

There is a lot in this paper. The Introduction is clear and persuasive that the general topic - how cancer associated fibroblasts (CAFs) are reprogrammed by cancer cells and how the CAFs then impact on cancer growth - is both fascinating, and a good route towards potential therapeutics; the data presented provides a good case for activin being a pivotal player in these interactions, at least for some skin cancers.

To summarise, the authors show that over expressing activin in cancer cells in vivo makes these cells much more invasive and triggers fibroblasts in their vicinity to be more fibrotic, and the authors also report similar in their organotypic co-culture studies; other in vitro studies show skin fibroblasts alone but exposed to activin also become more migratory and upregulate CAF marker genes.

In addition we are shown an elegant study in which fibroblasts overexpressing activin are allowed to deposit matrix and this matrix, as well as the captured "secretome" from these fibroblasts, leads to enhanced proliferation and invasive growth in cancer cells lines seeded onto it. It would be good to rule out (or not) that one component of this "cancer enhancing" matrix is not activin secreted by the fibroblasts.

Crossing a mouse line which spontaneously generates pre-cancerous papillomas with one overexpressing activin in keratinocytes provides a nice opportunity to characterise the transcriptome of activin-induced CAFs and this appears to overlap very well with other known CAF signatures, including a number of "cell migration" genes, such as mDia. We are told of the correlation in human cancers - hi activin in cancer cells associates with hi mDia in CAFs - and how shRNA knockdown of mDia in mouse or in human fibroblasts blocks activin-induced cell proliferation. They then show a direct regulatory link between mDia and various CAF marker genes via inhibition of p53 nuclear accumulation.

These basic science studies of how activin expression by cancer cells might drive CAF phenotype are taken through towards a potential lead for therapy by co-injection of activin-unresponsive fibroblasts and by forcing expression of the activin inhibitor follistatin in tumour cells where, in both cases, tumour growth in mice was delayed.

As I said at the beginning of this review - there is a lot here - and I would feel a little mean spirited to suggest further experiments. Those presented seem thorough and convincing that this pathway is indeed worth serious consideration as a target to inhibit progression in some skin cancers. Very nice work.

Referee #2 (Comments on Novelty/Model System for Author):

Some of the conclusions are based on data that is not statistically significant.

Referee #2 (Remarks for Author):

Review: A paracrine activin A-mDia2 axis promotes squamous carcinogenesis via fibroblast reprogramming

This is an interesting manuscript showing how SCC produce Activin A to control mDia in fibroblasts inducing their pro-tumorigenic activities. Authors are to be commended for the work, but there are still some issues that need to be addressed before publication.

1. Figure 1. Regarding the observation that fibroblasts are different if they are treated with Activin vs TGFbeta. What would happen if they are treated with both factors? This would probably better mirror the in vivo situation and maybe the resulting fibroblasts have combined synergistic properties.

2. Figure 5. Authors identify the markers of the CAFs (INHBA and mDia2) in cholangio and breast cancer but then they provide prognostic value in other tumours (ovarian, esophageal, melanoma) . Apart from colon cancer, are there any other tumour types in which they can find that CAF markers are upregulated and they confer poor prognosis within the same tumour type?

3. Figure 6. The p53 mechanistic connection is less convincing, wb show very weak signal for p53 making it difficult to quantify (F,G) and some of the data presented is not statistically significant (6G or MMP13 expression in H,I) What about the other genes identified in 6D? can authors stain for p53 in vivo in their tissue sections?

4. Figure 7: B: co-staining of mDia2 with a fibroblast marker would help (cells don't look like fibroblasts in the picture below). What about periostin, MMP13,etc protein expression? Are the fibroblasts reprogrammed? Why tumour weight is shown instead of tumour growth like in all the other tumour experiments. G: Why is this experiment represented in this manner instead of

previous representations of tumor growth? What about SCC13 tumors treated with FST? What about stroma with the treatment (CAF proportion/activation/mDia2 expression)? 7F is not statistically significant ($p=0.099$) so it is difficult to conclude anything from this data.

5. In vivo treatment with SMIFH2 would reinforce the conclusions since it would reprogram the fibroblast matrixome/secretome etc in vivo. The argument that some cancer cells increase invasion after SMIFH2 treatment (in discussion) does not seem to apply here since SCC don't increase their invasion after in vitro treatment (7F).

Referee #3 (Comments on Novelty/Model System for Author):

Technical Quality (inc. statistical analysis):

The data are generally clearly presented. The experiments are cleverly designed and for the most part well-controlled. The overall technical quality of this manuscript is good, but unfortunately, it is not high enough at this stage for me to recommend publication. On a positive note, the authors have included a comprehensive methods section (including extensive supplementary materials) that should enable other laboratories to reproduce the key findings. There is also a proper description of the statistical tests used for data analyses. However, the quality of the data is in some cases not good enough to support the conclusions made, which are either too strong or open to alternative explanations.

Novelty:

The findings in this manuscript are to my knowledge novel, as a role of a paracrine activin A-mDia2-p53 axis in the reprogramming of fibroblasts to a CAF phenotype has so far not been shown. The message of the manuscript is, overall, interesting, and expands our knowledge from previous work of the Werner lab about the roles of tumour-stroma crosstalk in the development and progression of squamous cell carcinoma.

Medical impact:

Soluble activin inhibitors are currently in clinical trials and show promising effects for the treatment of cancer-associated cachexia. But whether these inhibitors are also effective for anti-cancer treatment is not clear yet. Consequently the findings in this manuscript are interesting also from a clinical perspective.

Adequacy of model system:

The chosen model systems are adequate for this study. E.g. using NOD/SCID mice to study the roles of CAFs in SCC tumorigenesis, and using SCC13 cells which are only weakly tumorigenic, as well as more malignant A431 cells (although they are derived from a vulval carcinoma and not a cutaneous SCC).

Referee #3 (Remarks for Author):

This is an interesting manuscript highlighting the roles of a paracrine activinA-mDia2-p53 signalling axis in tumour-stroma crosstalk and fibroblast-CAF reprogramming in the context of squamous cell carcinoma.

The data are generally clearly presented. The experiments are cleverly designed and for the most part well-controlled. The overall technical quality of this manuscript is good, but unfortunately, it is not high enough at this stage for me to recommend publication. On a positive note, the authors have included a comprehensive methods section (including extensive supplementary materials) that should enable other laboratories to reproduce the key findings. There is also a proper description of the statistical tests used for data analyses. However, the quality of the data is in some cases not good enough to support the conclusions made, which are either too strong or open to alternative explanations.

Major concerns:

- Figs. 1, EV1 and EV2: I am not entirely convinced by the data showing that activin A expression in either SCC13 cells or fibroblasts is promoting tumour cell invasion. I'd therefore encourage the authors to address my concerns, mostly by re-analysing some of their existing data:

Fig. 1D: why were the data from the two SCC13-Act clones pooled in this graph?

Fig. 1E: penetration of ear cartilage tissue is an important evidence for activin A-enhanced cell invasion of xeno-transplanted SCC13 cells. Was this a statistically significant phenotype? Could the authors please include a higher magnification of the invaded cartilage?

Fig. 1F: what do the % in the graph refer to? E.g. percentage of Herovici-positive area per field of view? Is there an increased number of fibroblasts recruited in the vicinity of the tumour tissue? Was proliferation increased in the tumour tissue and/or the stroma?

In Fig. 1G and EV2A,B it is very difficult to see to what extent activation of activin A increases invasion into the dermal compartment, as EV-infected SCC13 cells are already invasive (see Berning et al. 2015 Tissue Engineering Part C). A quantification of dermal invasion would be very helpful. Maybe, instead of quantifying the total K14-stained area of the organotypic skin tissue, the authors could include a basement membrane marker in their stainings and quantify only the K14-stained tissue area that extends past the basement membrane? A quantification of proliferation markers would also be informative.

In my opinion, it is difficult to conclude from the organotypic cell culture data that the invasive growth of the SCC cells is mediated by the fibroblasts, as I assume the SCC cells would invade into any ECM substrate also in the absence of fibroblasts. Maybe the authors could address this?

Fig. EV2C: were similar changes in E-cadherin and beta catenin localisation observed with SCC13 cells?

Fig. EV2E and text on page 6: it is hard to see any difference in MMP expression between A431-EV and A431-Act lysates. Maybe this is because these cells are already very invasive? There appears to be a difference in the case of the SCC13 cells, but this should be quantified in order to support the statement in the text.

Fig. 3, panels D-H: what is the difference between the quantifications in panels D and F versus H. Does colony area refer to average area of individual colonies? The effect of co-culture is far less pronounced than that of conditioned medium or ECM. Could the authors comment on this? Which clone was used for the experiments? The quality of the immunostainings in panel E is poor.

Fig. 3, panel I: the quality of the H&E stainings is poor and precludes to see any evidence for increased invasion of SCC13 cells. I suggest therefore to perform similar quantifications on tumour cell invasion as mentioned above.

- The quality of some of the mechanistic and pre-clinical data that are supposed to provide evidence for the mDia2-p53 interaction and its roles in CAF marker expression and for blockade of activin A signalling as a potential strategy to inhibit SCC tumourigenesis, respectively, is unfortunately poor. The authors should therefore try to improve the quality of the data referred to below or tone down their interpretation in the text.

Fig. 6, panels F, G and text on page 12: increased/reduced p53 expression in the nuclear fractions is difficult to see in these representative western blots.

Panel I: percentage of cells with nuclear p53 should be quantified.

Panel J: is the difference in Postn expression between sh-EV and sh-EV+PFT really not statistically significant? On the other hand, MMP13 expression does not seem to differ significantly between any of the different conditions.

Fig. EV4N-P: there doesn't appear to be any knock-down of mDia2, especially under the empty vector conditions, making me question the results presented in panels O and P.

Fig. 7F: SMIFH2 treatment does not appear to have a significant effect, as otherwise stated in the text on page 14. The representative spheroid of ctrl cells at day 3 is difficult to make out in the image.

Minor concerns:

- Presentation of the data would benefit from slightly more comprehensive figure legends and improved labelling of axes in graphs and figures in general. E.g. please indicate more clearly which clones were used in the respective experiments, and which cell lines/cell types.
- Fig. 4A: I am wondering why the skin CAF signature was not positively enriched?
- Fig. 4H: what are the different stromal cell types expressing mDia2? Looks like strong mDIA2 expression on vessels.
- My personal preference is to show individual data points in cases of $N < 5$, but I am not aware of any related formatting requirements in the author guidelines.

1st Revision - authors' response

24th Dec 2019

Referee #1 (Comments on Novelty/Model System for Author):

Very thorough study combining mouse in vivo studies and in vitro studies with murine and human cells.

Referee #1 (Remarks for Author):

There is a lot in this paper. The Introduction is clear and persuasive that the general topic - how cancer associated fibroblasts (CAFs) are reprogrammed by cancer cells and how the CAFs then impact on cancer growth - is both fascinating, and a good route towards potential therapeutics; the data presented provides a good case for activin being a pivotal player in these interactions, at least for some skin cancers.

To summarise, the authors show that over expressing activin in cancer cells in vivo makes these cells much more invasive and triggers fibroblasts in their vicinity to be more fibrotic, and the authors also report similar in their organotypic co-culture studies; other in vitro studies show skin fibroblasts alone but exposed to activin also become more migratory and upregulate CAF marker genes.

In addition we are shown an elegant study in which fibroblasts overexpressing activin are allowed to deposit matrix and this matrix, as well as the captured "secretome" from these fibroblasts, leads to enhanced proliferation and invasive growth in cancer cells lines seeded onto it. It would be good to rule out (or not) that one component of this "cancer enhancing" matrix is not activin secreted by the fibroblasts.

Our reply: We thank the reviewer for these very positive comments. We have addressed the potential role of activin in the secretome using a tumor spheroid assay. When we treated the SCC13 spheroids with

conditioned medium of fb Act or fb EV, the conditioned medium of activin A overexpressing cells strongly promoted spheroid formation compared to the medium of control cells. By contrast, addition of recombinant activin A to the spheroids had no effect. These results strongly suggest that factors induced by activin A are responsible for the tumor-promoting effect of activin-exposed fibroblasts rather than activin A itself. This result is now shown in Fig 3I.

Crossing a mouse line which spontaneously generates pre-cancerous papillomas with one overexpressing activin in keratinocytes provides a nice opportunity to characterise the transcriptome of activin-induced CAFs and this appears to overlap very well with other known CAF signatures, including a number of "cell migration" genes, such as mDia. We are told of the correlation in human cancers - hi activin in cancer cells associates with hi mDia in CAFs - and how shRNA knockdown of mDia in mouse or in human fibroblasts blocks activin-induced cell proliferation. They then show a direct regulatory link between mDia and various CAF marker genes via inhibition of p53 nuclear accumulation.

These basic science studies of how activin expression by cancer cells might drive CAF phenotype are taken through towards a potential lead for therapy by co-injection of activin-unresponsive fibroblasts and by forcing expression of the activin inhibitor follistatin in tumour cells where, in both cases, tumour growth in mice was delayed.

As I said at the beginning of this review - there is a lot here - and I would feel a little mean spirited to suggest further experiments. Those presented seem thorough and convincing that this pathway is indeed worth serious consideration as a target to inhibit progression in some skin cancers. Very nice work.

Referee #2 (Comments on Novelty/Model System for Author):

Some of the conclusions are based on data that is not statistically significant.

Referee #2 (Remarks for Author):

Review: A paracrine activin A-mDia2 axis promotes squamous carcinogenesis via fibroblast reprogramming

This is an interesting manuscript showing how SCC produce Activin A to control mDia in fibroblasts inducing their pro-tumorigenic activities. Authors are to be commended for the work, but there are still some issues that need to be addressed before publication.

1. Figure 1. Regarding the observation that fibroblasts are different if they are treated with Activin vs TGFbeta. What would happen if they are treated with

both factors? This would probably better mirror the in vivo situation and maybe the resulting fibroblasts have combined synergistic properties.

Our reply: In response to this interesting question, we treated fibroblasts with both activin A and TGF- β 1 at concentrations that we and others had shown to promote collagen gel contraction by fibroblasts. Interestingly, we observed that activin A inhibited TGF- β 1-induced gel contraction under our experimental conditions (see Fig 2B, new version). These results further demonstrate that activin A and TGF- β 1 have non-overlapping effects in our experimental systems (see also Fig 2D-F). Given that we employed physiologically relevant activin A concentrations, these observations suggest that the activity of activin A is dominant. In the future it will be important to perform dose-response experiments varying one factor at a time as well as to add activin A or TGF- β 1 at different times. However, we believe that this is beyond the focus of the present study, which is the activin A-mDia2 axis.

2. Figure 5. Authors identify the markers of the CAFs (INHBA and mDia2) in cholangio and breast cancer but then they provide prognostic value in other tumours (ovarian, esophageal, melanoma). Apart from colon cancer, are there any other tumour types in which they can find that CAF markers are upregulated and they confer poor prognosis within the same tumour type?

Our reply: Metanalyses suggest that high *INHBA* and *DIAPH3* expression also correlates significantly with poor prognosis in liver, stomach and breast cancers. These new data are now shown in Fig 5H instead of the data from the other tumors, which we showed in the initial version. To be noted, we have now performed survival analyses using KMPlotter, which is a more stringent and comprehensive tool that analyzes gene expression data and overall survival across large datasets from GEO, EGA and TCGA (doi: 10.1038/s41598-018-27521-y.). The data shown in the initial version of our manuscript had been obtained using the Proggene tool, which only analyzes overall survival from a single dataset (lower patient numbers). Furthermore, the online access of Proggene is not publicly available anymore. We have modified the figure legend and the paragraph in Materials and Methods accordingly.

3. Figure 6. The p53 mechanistic connection is less convincing, we show very weak signal for p53 making it difficult to quantify (F,G) and some of the data presented is not statistically significant (6G or MMP13 expression in H,I) What about the other genes identified in 6D? can authors stain for p53 in vivo in their tissue sections?

Our reply: As requested, we repeated the experiments using a new p53 antibody, and we now combine data obtained with both antibodies and present improved Western blots (see Fig 6F and G, new version). The antibody information has been added to Supplementary Materials and Methods (see Appendix). We reproduced the decrease in nuclear p53

upon activin A treatment of fibroblasts using this new antibody, and the difference is now statistically different.

Furthermore, we repeated the experiments shown in Fig 6J using sh-EV and sh-mDia2 fibroblasts from early passage (passage 3-4) and we achieved a better mDia2 knock-down efficiency. We also included additional genes in the analysis, including genes identified in the experiments shown in Fig 6D, such as *Fn1*, *Acta2* and *Spp1*. Importantly, expression of all these genes was significantly down-regulated in sh-mDia2 fibroblasts when compared to sh-EV controls. Furthermore, addition of PFT α to sh-mDia2 fibroblasts partially rescued the down-regulation of these CAF genes.

As suggested by the reviewer, we stained sections from tumors formed by SCC13 Act cells in the presence of control fibroblasts or fibroblasts with mDia2 knock-down with antibodies against periostin or p53. mDia2 knock-down in fibroblasts indeed resulted in a strong reduction in periostin levels in the stroma. In addition, p53 puncta were readily visible in the nuclei of stromal cells. These data are now shown in Fig 6M and N.

4. *Figure 7: B: co-staining of mDia2 with a fibroblast marker would help (cells don't look like fibroblasts in the picture below). What about periostin, MMP13, etc protein expression? Are the fibroblasts reprogrammed?*

Our reply: Cell shape in tissue sections is influenced by several factors, including the way cells are cut. For example, the cells showing high mDia2 levels in the top right corner of the lower panel in Fig 7B (K14/mDia2/Hoechst micrographs) strongly resemble those depicted in the top panel. To go beyond morphological analyses, we performed co-staining for mDia2 and the pan-fibroblast marker PDGFR- α and found a strong co-localization of both markers. Therefore, a large percentage of the mDia2-positive cells in the stroma are indeed fibroblasts. These new data are now shown in Fig 7B. Together with the p53 localization data and the down-regulation of periostin in the tumor stroma (as addressed above in comment No.3, Fig 6M and 6N), these data support the notion that fibroblasts indeed undergo reprogramming *in vivo* after mDia2 knock-down.

Why tumour weight is shown instead of tumour growth like in all the other tumour experiments. G: Why is this experiment represented in this manner instead of previous representations of tumor growth?

Our reply: In the absence of fibroblasts, tumor formation by SCC13 cells is very slow and takes several weeks. Since measurements of tumor volume require anaesthesia of the mice, we could not do this so frequently and for such a long time period. Therefore, we only looked at the end point in this experiment (Fig 7C). We believe that this is acceptable, since tumor weight and tumor volume show a direct correlation in all our experiments (see for example Fig 1C and D). Of

note, this is usually the case for solid tumors. We have clarified this further in the text (page 14).

In Fig 7G, we started the treatment when tumors had already formed and we wanted to compare the change in the tumor volume between the mock- and the FST-treated groups. In response to this comment, we now present the data as in the previous experiments. We also clarify in the legend that day 0 is the time when the tumors were first injected with follistatin (3-4 weeks after injection of the cells).

What about SCC13 tumors treated with FST? What about stroma with the treatment (CAF proportion/activation/mDia2 expression)? 7F is not statistically significant ($p=0.099$) so it is difficult to conclude anything from this data.

Our reply: As mentioned above, SCC13 tumors grow very slowly in the absence of fibroblasts. Analysis of FST in this experiment would have required a large number of injections over a long time period. Therefore, we have not performed this experiment for animal welfare reasons. However, we show the effect of FST on SCC13 tumors *ex vivo* in Fig 7E. As requested by the reviewer, we stained sections from the FST-treated tumors for mDia2 and found reduced expression of mDia2 in the stroma of the FST-treated tumors. A representative staining was added to Fig 7G. We repeated the experiment shown in Fig 7F. Upon analysis of all independent replicates the difference was statistically significant.

5. In vivo treatment with SMIFH2 would reinforce the conclusions since it would reprogram the fibroblast matrisome/secretome etc in vivo. The argument that some cancer cells increase invasion after SMIFH2 treatment (in discussion) does not seem to apply here since SCC don't increase their invasion after in vitro treatment (7F).

Our reply: We agree that this would be an interesting experiment. Unfortunately, however, the maximal tolerable dose for *in vivo* studies, as well as the pharmacokinetics and pharmacodynamics of SMIFH2 have not yet been established in mice, making it difficult to conduct meaningful intervention studies. In one study, injection of SMIFH2 into zebrafish larvae induced developmental defects (doi: [10.3389/fphar.2018.00340](https://doi.org/10.3389/fphar.2018.00340)). Nevertheless, we performed this experiment and we observed a mild, but non-significant reduction in tumor size in SMIFH2-treated A431 tumors at day 9 post-treatment. Unfortunately, we had to terminate the experiment for animal welfare reasons, due to toxicity of the administered SMIFH2 dose. Indeed, the SMIFH2 treatment seemed to affect the overall well-being of the mice. Scabs were visible around the SMIFH2-treated A431 tumors, but not in controls. Furthermore, the ears displayed reduced elasticity and the skin became fragile. We show a figure of these results below (not for publication) for the information of the reviewer (note that data are plotted as in Fig 7G).

Referee #3 (Comments on Novelty/Model System for Author):

Technical Quality (inc. statistical analysis):

The data are generally clearly presented. The experiments are cleverly designed and for the most part well-controlled. The overall technical quality of this manuscript is good, but unfortunately, it is not high enough at this stage for me to recommend publication. On a positive note, the authors have included a comprehensive methods section (including extensive supplementary materials) that should enable other laboratories to reproduce the key findings. There is also a proper description of the statistical tests used for data analyses. However, the quality of the data is in some cases not good enough to support the conclusions made, which are either too strong or open to alternative explanations.

Novelty:

The findings in this manuscript are to my knowledge novel, as a role of a paracrine activin A-mDia2-p53 axis in the reprogramming of fibroblasts to a CAF phenotype has so far not been shown. The message of the manuscript is, overall, interesting, and expands our knowledge from previous work of the Werner lab about the roles of tumour-stroma crosstalk in the development and progression of squamous cell carcinoma.

Medical impact:

Soluble activin inhibitors are currently in clinical trials and show promising effects for the treatment of cancer-associated cachexia. But whether these inhibitors are also effective for anti-cancer treatment is not clear yet. Consequently, the findings in this manuscript are interesting also from a clinical perspective.

Adequacy of model system:

The chosen model systems are adequate for this study. E.g. using NOD/SCID mice to study the roles of CAFs in SCC tumorigenesis, and using SCC13 cells which are

only weakly tumourigenic, as well as more malignant A431 cells (although they are derived from a vulvar carcinoma and not a cutaneous SCC).

Our reply: We thank the reviewer for the positive comments. We are aware that A431 cells are derived from a vulvar carcinoma and we now mention this explicitly in the next (page 5). However, our data show that the results are relevant for different types of SCCs and also for other cancers and therefore, the use of A431 cells strengthens the general relevance of our data.

Referee #3 (Remarks for Author):

This is an interesting manuscript highlighting the roles of a paracrine activinA-mDia2-p53 signalling axis in tumour-stroma crosstalk and fibroblast-CAF reprogramming in the context fo squamous cell carcinoma.

The data are generally clearly presented. The experiments are cleverly designed and for the most part well-controlled. The overall technical quality of this manuscript is good, but unfortunately, it is not high enough at this stage for me to recommend publication. On a positive note, the authors have included a comprehensive methods section (including extensive supplementary materials) that should enable other laboratories to reproduce the key findings. There is also a proper description of the statistical tests used for data analyses. However, the quality of the data is in some cases not good enough to support the conclusions made, which are either too strong or open to alternative explanations.

Major concerns:

Figs. 1, EV1 and EV2: I am not entirely convinced by the data showing that activin A expression in either SCC13 cells or fibroblasts is promoting tumour cell invasion. I'd therefore encourage the authors to address my concerns, mostly by re-analysing some of their existing data:

Our reply: As requested by the reviewer, we re-analyzed some of our data and we performed additional experiments to address the concerns of the reviewer.

Fig. 1D: why were the data from the two SCC13-Act clones pooled in this graph?

Our reply: We initially pooled them, but showed the results with the two clones in different colors. As requested, we now show them independently (Fig 1E).

Fig. 1E: penetration of ear cartilage tissue is an important evidence for activin A-enhanced cell invasion of xeno-transplanted SCC13 cells. Was this a statistically significant phenotype? Could the authors please include a higher magnification of the invaded cartilage?

Our reply: In response to this question, the sections were evaluated by two independent blinded investigators and classified as invasive or non-

invasive. Quantitative analysis of these results showed that tumors formed by SCC13 cells overexpressing activin A are significantly more invasive when compared to the control group (Fig 1D). In addition, we included a higher magnification of the area where the tumor cells invade into the cartilage (Fig 1D, new version with improved resolution).

Fig. 1F: what do the % in the graph refer to? E.g. percentage of Herovici-positive area per field of view? Is there an increased number of fibroblasts recruited in the vicinity of the tumour tissue? Was proliferation increased in the tumour tissue and/or the stroma?

Our reply: Yes, the graph represents percentage of Herovici-positive area per field of view. We have clarified this in the figure (y-axis) and in the legend.

As requested by the reviewer, we co-stained the sections with antibodies for the fibroblast marker PDGFR- α and the proliferation marker Ki67. The data demonstrate that activin A overexpression in SCC13 cells indeed causes a significant increase in the number of proliferating fibroblasts in the tumor stroma. These new data are now shown in Fig 1G.

In Fig. 1G and EV2A,B it is very difficult to see to what extent activation of activin A increases invasion into the dermal compartment, as EV-infected SCC13 cells are already invasive (see Berning et al. 2015 Tissue Engineering Part C). A quantification of dermal invasion would be very helpful. Maybe, instead of quantifying the total K14-stained area of the organotypic skin tissue, the authors could include a basement membrane marker in their stainings and quantify only the K14-stained tissue area that extends past the basement membrane? A quantification of proliferation markers would also be informative.

Our reply: As suggested by the reviewer, we stained the sections with antibodies for the basement membrane marker collagen IV, and we quantified the K14-positive area below the basement membrane. These new results verify the increased invasion by SCC13 cells overexpressing activin A when compared to EV controls and are shown in Fig 1J. Note that this increase in invasiveness is associated by a severe reduction in the integrity of the basement membrane. This observation is in line with the increased MMP release and activity (see EV2E). Furthermore, the sections were stained for Ki67 and the quantification of this result is shown in Fig 1I.

In my opinion, it is difficult to conclude from the organotypic cell culture data that the invasive growth of the SCC cells is mediated by the fibroblasts, as I assume the SCC cells would invade into any ECM substrate also in the absence of fibroblasts. Maybe the authors could address this?

Our reply: We addressed this question by seeding SCC13 tumor spheroids onto a collagen gel with or without incorporated fibroblasts and we found that SCC13 cells only invaded into the gel in the presence of

fibroblasts. We show a figure of these results below for the information of the reviewer.

Fig. EV2C: were similar changes in E-cadherin and beta catenin localisation observed with SCC13 cells?

Our reply: Yes, similar changes were seen with SCC13 cells, and we now show a higher magnification of the E-cadherin and β -catenin staining in Fig EV1C.

Fig. EV2E and text on page 6: it is hard to see any difference in MMP expression between A431-EV and A431-Act lysates. Maybe this is because these cells are already very invasive? There appears to be a difference in the case of the SCC13 cells, but this should be quantified in order to support the statement in the text.

Our reply: As requested by the reviewer we quantified these data and the quantification is now shown in the Fig EV2E.

Fig. 3, panels D-H: what is the difference between the quantifications in panels D and F versus H. Does colony area refer to average area of individual colonies? The effect of co-culture is far less pronounced than that of conditioned medium or ECM. Could the authors comment on this? Which clone was used for the experiments? The quality of the immunostainings in panel E is poor.

Our reply: Panels D and F show the effect of conditioned medium or extracellular matrix on colony formation and panel H shows the effect of the co-culture. The graphs show the area covered by all colonies per area. This is now clarified in the legend.

Indeed, the effect of the co-culture is less pronounced, since ECM and conditioned medium were collected for three days and the conditioned medium of the culture was concentrated. We now clarify this issue in the text (page 8).

We also mention in the legend that clone 2 was used for the experiment. The quality of the immunostaining in panel E was improved (Fig 3E).

Fig. 3, panel I: the quality of the H&E stainings is poor and precludes to see any

evidence for increased invasion of SCC13 cells. I suggest therefore to perform similar quantifications on tumour cell invasion as mentioned above.

Our reply: As suggested by the reviewer, we replaced the H&E staining by K14 immunofluorescence staining. The area of K14-positive cells was quantified (Fig 3J, new version). We would like to point out that we focused here on the increased thickness of the epithelium as a result of cancer cell proliferation rather than on its invasiveness.

The quality of some of the mechanistic and pre-clinical data that are supposed to provide evidence for the mDia2-p53 interaction and its roles in CAF marker expression and for blockade of activin A signalling as a potential strategy to inhibit SCC tumourigenesis, respectively, is unfortunately poor. The authors should therefore try to improve the quality of the data referred to below or tone down their interpretation in the text.

Our reply: As requested by this reviewer and also by reviewer 2, we repeated some of the p53 experiments and the quality of the data has been improved.

Fig. 6, panels F, G and text on page 12: increased/reduced p53 expression in the nuclear fractions is difficult to see in these representative western blots.

Our reply: The Western blots were repeated with another p53 antibody, and we replaced the previous blot by a new blot obtained with this new antibody.

Panel I: percentage of cells with nuclear p53 should be quantified.

Our reply: We quantified the percentage of cells with nuclear p53 staining and the data are now shown in Fig 6I. The difference between vehicle and PFT α -treated cells is statistically significant.

Panel J: is the difference in Postn expression between sh-EV and sh-EV+PFT really not statistically significant? On the other hand, MMP13 expression does not seem to differ significantly between any of the different conditions.

Our reply: As requested by the reviewer and also by reviewer 2, we repeated the experiments and we believe that the quality of the data has been improved. There is also a clear difference in MMP13 expression.

Fig. EV4N-P: there doesn't appear to be any knock-down of mDia2, especially under the empty vector conditions, making me question the results presented in panels O and P.

Our reply: We agree that the knock-down did not look convincing in this figure. We therefore repeated this experiment and the new data (with a much more efficient mDia2 knock-down) are shown in Fig EV4C.

Fig. 7F: SMIFH2 treatment does not appear to have a significant effect, as otherwise stated in the text on page 14. The representative spheroid of ctrl cells at day 3 is difficult to make out in the image.

Our reply: We repeated the experiments to increase the sample size and the difference is now statistically significant. The figure has been replaced by a new figure (Fig 7F), which also includes slightly contrasted photomicrographs of the spheroids that allow the reader to better identify the spheroids' margins.

Minor concerns:

- *Presentation of the data would benefit from slightly more comprehensive figure legends and improved labelling of axes in graphs and figures in general. E.g. please indicate more clearly which clones were used in the respective experiments, and which cell lines/cell types.*

Our reply: We have improved the labeling of the figures and extended the figure legends. In particular, we mention the clone number in all legends.

- *Fig. 4A: I am wondering why the skin CAF signature was not positively enriched?*

Our reply: According to the information from GSE37738, these fibroblasts were not directly sorted for microarray analysis, but initially cultured from primary tumors. Hence, the cultured fibroblasts might lose their pro-tumorigenic characteristics. Importantly, direct comparison to cultured RDEB-SCC CAFs (also from the same dataset, GSE37738) showed higher enrichment of the skin CAF signature (NES score 1.55). This suggests that the transcriptomic signature of activin A-exposed fibroblasts positively correlates with the signature of more aggressive cutaneous SCCs. We have clarified this further in the text (page 9).

- *Fig. 4H: what are the different stromal cell types expressing mDia2? Looks like strong mDIA2 expression on vessels.*

Our reply: To address this question, we performed co-immunofluorescence staining using antibodies against mDia2 and the blood or lymphatic endothelial cell markers MECA-32 or LYVE1, respectively. Indeed, we observed co-localization of mDia2 and MECA32, but not of mDia2 and LYVE1 in tumor sections, demonstrating that blood vessel endothelial cells in these tumors are mDia2 positive. In addition, other stromal cells express mDia2 (see Fig 4H, new version). The latter are most likely fibroblasts as demonstrated by the result shown in Fig 7B.

- *My personal preference is to show individual data points in cases of $N < 5$, but I am not aware of any related formatting requirements in the author guidelines.*

Our reply: There are indeed no such formatting requirements in the authors guidelines and therefore, we prefer to show all data in the same style. However, we show the original values and exact p-values in the Supplement.

2nd Editorial Decision

30th Jan 2020

Thank you for the submission of your revised manuscript to EMBO Molecular Medicine, and my apologies for the delay in getting back to you following the holiday season. We have now received the enclosed reports from the three referees who had originally reviewed your manuscript. As you will see, they are all supportive of publication, and I am thus pleased to inform you that we will be able to accept your manuscript pending the following final editorial amendments.

***** Reviewer's comments *****

Referee #1 (Comments on Novelty/Model System for Author):

As you know I liked the paper on first submission and felt it needed very little additional work. I've read through the revised MS and the authors rebuttal letter outlining how they have dealt with all of our concerns. It seems that the authors have addressed all criticisms and so, in my view the paper is now acceptable for publication as is.

Referee #2 (Remarks for Author):

The authors have addressed all of my concerns.

Referee #3 (Comments on Novelty/Model System for Author):

The technical quality has been significantly improved and the manuscript is now publication-ready.

Referee #3 (Remarks for Author):

The authors have satisfactorily addressed all my comments. I congratulate the authors to an exciting piece of work!

2nd Revision - authors' response

6th Feb 2020

The authors performed the requested editorial changes.

Corresponding Author Name: Sabine Werner and Michael Cangkrama

Manuscript Number: EMM-2019-11466